# A critical period of translational control during brain development at codon resolution

Dermot Harnett [1,11] ✉, Mateusz C. Ambrozkiewicz [2,11] ✉, Ulrike Zinnall [1], Alexandra Rusanova[3], Ekaterina Borisova[3], Amelie N. Drescher[4], Marta Couce-Iglesias[4], Gabriel Villamil [1], Rike Dannenberg[2], Koshi Imami [5,6], Agnieszka Münster-Wandowski[7], Beatrix Fauler[4], Thorsten Mielke [4], Matthias Selbach [6], Markus Landthaler [1,8], Christian M. T. Spahn[9], Victor Tarabykin[2,3,12], Uwe Ohler [1,8,10,12] & Matthew L. Kraushar [4,12] ✉

Translation modulates the timing and amplification of gene expression after transcription. Brain development requires uniquely complex gene expression patterns, but large-scale measurements of translation directly in the prenatal brain are lacking. We measure the reactants, synthesis and products of mRNA translation spanning mouse neocortex neurogenesis, and discover a transient window of dynamic regulation at mid-gestation. Timed translation upregulation of chromatin-binding proteins like Satb2, which is essential for neuronal subtype differentiation, restricts protein expression in neuronal lineages despite broad transcriptional priming in progenitors. In contrast, translation downregulation of ribosomal proteins sharply decreases ribosome biogenesis, coinciding with a major shift in protein synthesis dynamics at mid-gestation. Changing activity of eIF4EBP1, a direct inhibitor of ribosome biogenesis, is concurrent with ribosome downregulation and affects neurogenesis of the Satb2 lineage. Thus, the molecular logic of brain development includes the refinement of transcriptional programs by translation. Modeling of the developmental neocortex translatome is provided as an open-source searchable resource at https://shiny.mdc-berlin.de/cortexomics.

Changes in translation activity can lead to substantial discrepancies between mRNA and protein for the same gene, and are a hallmark of many dynamic cellular transition states[1]. Cellular transitions are uniquely complex during prenatal brain development, when neural stem cells deploy highly sophisticated gene expression programs for neuronal specification[2,3]. In evolutionarily advanced brain regions like the neocortex, a cell's transcriptional signature alone appears insufficient to account for the enormous cellular diversity, and single-cell RNA sequencing (scRNA-seq) analyses support this idea[4–9]. Although transcriptomes define broad classes of neurons, a striking conclusion from recent studies is the degree of homogeneity in mRNA pools between distinct neuronal lineages during prenatal differentiation[4,8], in postnatal circuits[7], and even between neurons and astrocytes[10]. Considering whether neuronal differentiation in the neocortex uses a more 'generic' transcriptome[4] has led researchers in the field to ask recently whether neuronal identity is a stochastic rather than deterministic

A full list of affiliations appears at the end of the paper. ✉e-mail: dermot.p.harnett@gmail.com; mateusz-cyryl.ambrozkiewicz@charite.de; matthew.kraushar@molgen.mpg.de

process[11]. Do progenitors 'play dice'[12] while deciding their neuronal fate? Thus, the blueprint of gene expression in evolutionarily advanced brain regions is likely a multilayered, progressive refinement, including and beyond transcription[3]. Neocortex development may thus require particularly dynamic translational control[13,14].

Direct measurements of protein synthesis would provide a clearer picture of functional gene expression in the developing brain; however, a large-scale high-resolution analysis of mRNA translation during neurogenesis has lagged behind transcriptome analysis, in part owing to technical limitations in protein measurement. Targeted and selective protein synthesis refines the output of gene expression in brain development[5,15–19]. Abnormal ribosome levels and disrupted translation were found recently to be mid-gestation etiologies of neurodevelopmental disorders[20]. However, how ribosomes decode mRNA in the transcriptome-to-proteome transition during developmental neurogenesis remains unknown.

To circumvent these challenges and measure the temporal dynamics of the reactants, synthesis and products of mRNA translation during brain development, we performed sequencing of ribosome-protected mRNA fragments (Ribo-seq; ribosome profiling)[21] in parallel with RNA-seq, transfer RNA quantitative PCR (qPCR) array and mass spectrometry across five stages of mouse neocortex neurogenesis. By capturing ribosome–mRNA interactions at codon-level resolution, we find that ~18% of mRNAs change translation efficiency in the progressive specification of neural stem cells to post-mitotic neurons, with a transient peak window of dynamic translation at mid-gestation. We focus on the divergent cellular pathways most affected by translation upregulation vs downregulation, which include chromatin-binding proteins like *Satb2* (refs. [22,23]), and ribosomal proteins, respectively. An acute decrease in ribosome biogenesis coincides with widespread changes in global translation activity at mid-gestation. Finally, we investigate how a regulator of ribosomal protein translation, eIF4EBP1 (refs. [24–26]), affects neurogenesis of the Satb2 lineage. We provide the developmental neocortex translatome as an open-source searchable web resource at https://shiny.mdc-berlin.de/cortexomics.

## Results

### Translation regulation peaks at mid-neurogenesis

We focused our study on the mammalian neocortex, an evolutionarily advanced developmental system with a tightly timed sequence of neurogenesis[2,27] (Fig. 1a). At approximately embryonic day 12.5 (E12.5), this predominantly stem cell tissue gives birth to its first neurons. Neurons born early at E12.5 form distinct connections and control different functions than those born later at E15.5. By postnatal day 0 (P0), neurogenesis is largely complete. The timed sequence of gene expression is essential to specify neuronal fate from the stem cell pool.

Our experimental strategy is shown in Fig. 1a. Ribo-seq measures 80S ribosomes bound to the open reading frame (ORF) of mRNA, a quantitative indicator of active protein synthesis at codon-level resolution[21]. Our optimizations with neocortex lysates circumvented the requirement for pharmacological ribosome stalling with cycloheximide[18], which may introduce ribosome footprint redistribution artifacts[28], and enabled efficient nuclease digestion to generate high-fidelity ribosome-protected mRNA fragments (RPFs) (Extended Data Fig. 1). We obtained mRNA transcripts per million (TPM) and RPF densities for 22,373 genes (Extended Data Fig. 2a and Supplementary Table 1). Reproducibility of both mRNA and RPF measurements permitted reliable calculation of mRNA translation efficiency (Extended Data Fig. 2b–e), the ratio between ribosome binding to an mRNA's coding sequence and the mRNA's level overall (Fig. 1b). As a quality control, further analysis included coding sequences with 32 or more Ribo-seq footprints in at least one stage as per ref. [29], which resulted in a set of 12,228 translated GENCODE-annotated transcripts.

We first calculated fold changes between sequential time points in mRNA or RPF vs protein (Fig. 1c and Supplementary Table 2). Whereas

gene expression overall is quite stable between E12.5 and E14, a burst of regulation occurs at E15.5 at both transcriptional and translational levels, with a significant effect on the proteome. Calculation of translation efficiency highlighted a transient window of robust regulation at E15.5, coinciding with the major transition in neuronal fate specification. Translation efficiency upregulation was found to occur in 1,129 genes and downregulation was found to occur in 1,131 genes. A further 2,253 genes change in steady-state mRNA only, without any significant translation efficiency change. Therefore, we estimate that ~18% of the neocortex transcriptome is dynamically translated across neocortex neurogenesis, with an inflection point at mid-neurogenesis.

Our Ribo-seq data show a higher correlation with protein-level changes than RNA-seq data (Fig. 1c). We decomposed technical and systematic variation in protein levels, and estimated proportions explained by RNA-seq vs Ribo-seq[30] (Fig. 1d and Extended Data Fig. 2f). A majority of protein-level variance is accounted for by RNA-seq, in agreement with prior observations[30,31]. However, Ribo-seq as a measure of synthesis consistently explains a higher fraction of protein variation than RNA-seq, especially for proteins with increasing levels, and for mRNAs with changing translation efficiency.

### Translation upregulation of chromatin-binding protein Satb2

We first focused on the cohort of genes that have translation efficiency upregulation after E12.5 (Supplementary Table 2). Gene Ontology (GO) analysis demonstrated that chromatin-binding proteins are particularly subject to translation upregulation (Fig. 2a). Chromatin-binding proteins like transcription factors have a powerful influence on the neuronal fate of stem cells, which is tightly coordinated in developmental time. Early-born post-mitotic neurons ultimately express transcription factors like Bcl11b, which drives them to connect subcortically[32]. In contrast, late-born post-mitotic neurons after E15.5 ultimately express transcription factors like Satb2, which drives them to connect intracortically[22,23,33]. How proteins like transcription factors achieve neuronal subtype and temporally restricted expression is a critical unresolved question.

Among the most translationally upregulated neurodevelopmental proteins discovered in our data is the essential, late-born, upper-layer neuron transcription factor Satb2 (Fig. 2b). We assessed the trajectory of Satb2 synthesis in our RNA-seq, Ribo-seq and mass spectrometry data, along with calculated translation efficiency, in comparison with the intermediate filament protein Nes expressed by neural stem cells[34], and early-stage transcription factor Bcl11b expressed in neurons positioned adjacent to the later Satb2 lineage. As expected, Nes demonstrates predominantly transcriptionally driven expression downregulation, as the neural stem cell pool is depleted by neuronal differentiation[35]. Bcl11b is expressed in the early-born lineage with high concordance between RNA-seq and Ribo-seq, and with low fluctuations in translation efficiency. In contrast, from E14 to E15.5, the *Satb2* Ribo-seq signal increases 7.4-fold and the mass spectrometry signal increases 8.2-fold, in excess of the 5.4-fold change in RNA-seq, yielding a 1.4-fold increase in translation efficiency between these developmental stages. These data suggest that *Satb2* expression is amplified by translation.

To begin testing the hypothesis that *Satb2* mRNA undergoes translation regulation, we examined the cellular distribution of *Satb2* mRNA in scRNA-seq neuronal lineage-tracing data[4]. Surprisingly, we found that *Satb2* mRNA is robustly expressed in differentiated neurons of both the early-born and late-born lineages (Fig. 2c), an apparent discrepancy with previous findings for Satb2 protein[22,33]. Thus, transcription of this upper-layer program may occur in neuronal lineages that include lower layers, and outside of the expected protein distribution.

To directly visualize the spatiotemporal expression of *Satb2* mRNA and protein, we performed fluorescence in situ hybridization (FISH) and immunohistochemistry (IHC) in neocortex coronal sections (Fig. 2d), with probe and antibody specificity confirmed in *Satb2*[−/−] brains (Extended Data Fig. 3a), and signal quantified per cell (Fig. 2e,

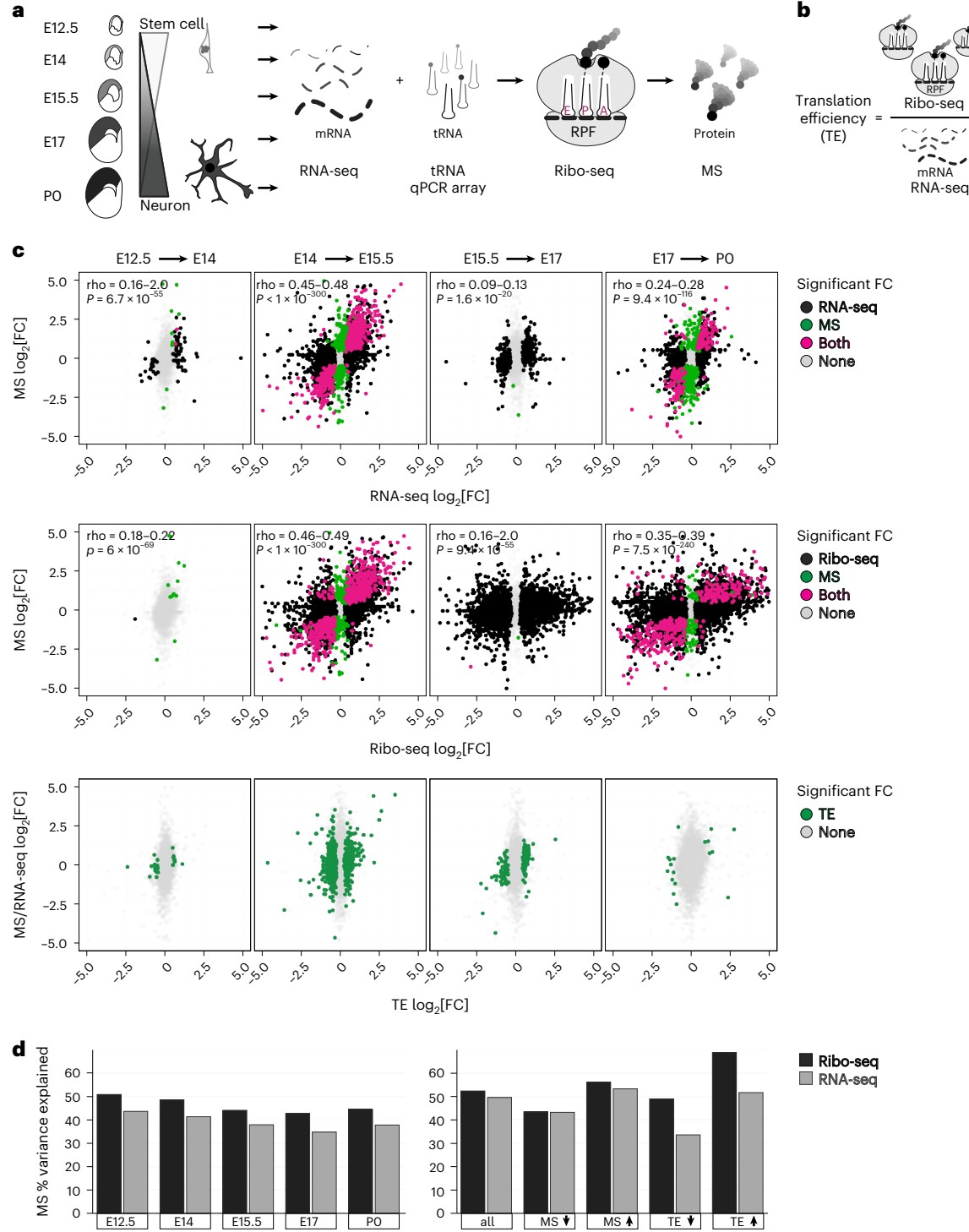

**Fig. 1 | A transient spike in translation regulation occurs at mid-neurogenesis during prenatal development. a**, Neural stem cell differentiation in the brain's neocortex, analyzed by RNA-seq, Ribo-seq, tRNA qPCR array and mass spectrometry (MS) at embryonic (E12.5–E17) and postnatal (P0) stages. **b**, Schematic of translation efficiency (TE). **c**, Sequential fold changes (FCs) in post-transcriptional gene expression between adjacent stages, comparing mRNA vs protein (top), mRNA translation vs protein (middle), and calculated translation efficiency (bottom). Differential expression was called with an empirical Bayes moderated two-sided *t*-test with adjustment for multiple comparisons. Significance assessed at ≥1.25 fold change, *P* < 0.05. **d**, The percent variance in mass spectrometry explained by RNA-seq or Ribo-seq at each developmental stage, and for subgroups with mass spectrometry and translation efficiency changes. See also Extended Data Figs. 1 and 2 and Supplementary Tables 1 and 2.

Extended Data Fig. 3b and Supplementary Table 3). At the onset of neurogenesis E12.5, initial scattered, weak Bcl11b protein signal is congruent with its mRNA signal in post-mitotic neurons. Satb2 protein is undetectable; however, we observed robust *Satb2* mRNA signal throughout the neocortex, from the ventricular zone in multipotent progenitors and throughout the nascent cortical plate in early-born post-mitotic neurons. In neurons, almost half of all *Satb2* mRNA clusters colocalize with *Bcl11b* mRNA, which rarely occurs in the stem cell niche.

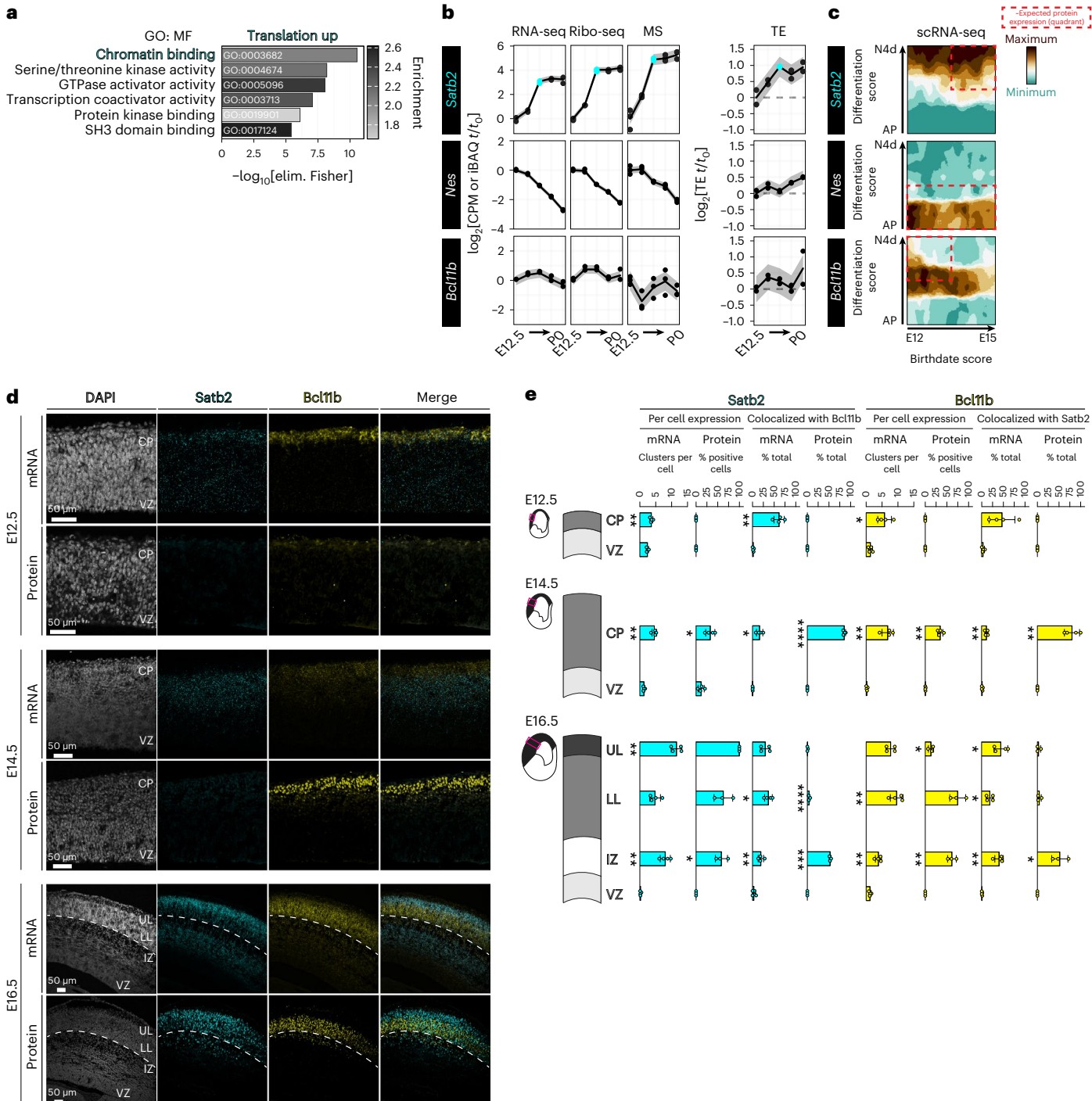

**Fig. 2 | Translation upregulation of *Satb2* leads to divergent spatiotemporal mRNA and protein expression. a**, GO (molecular function, MF) analysis of translationally upregulated (TE up) mRNAs. **b**, The median trajectory of *Satb2*, *Nes* and *Bcl11b* gene expression measured by RNA-seq, Ribo-seq, mass spectrometry and translation efficiency. The E15.5 time point is highlighted for *Satb2*. **c**, *Satb2*, *Nes* and *Bcl11b* expression in scRNA-seq data tracking differentiating neocortex cells from 1 h (apical progenitor, AP) to 96 h (neuron 4 days old, N4d) after birth (*y* axis), at birthdates E12, E13, E14 or E15 (*x* axis), with expression levels calculated in ref. [4]. Expected distribution of protein expression[2] is outlined. **d**, Neocortex coronal sections at E12.5, E14.5 and E16.5

analyzed for *Satb2* and *Bcl11b* mRNA by FISH, and protein by IHC. Deep border of the cortical plate is demarcated at E16.5 (dotted line). Nuclei are stained with DAPI. CP, cortical plate; VZ, ventricular zone; UL, upper layers; LL, lower layers. **e**, Quantification of **d**; *n* = 3 independent brains for E16.5 IHC, *n* = 4 independent brains for all other stages or assays. Mean ± s.d. is shown. Comparison of adjacent cortical layers starting from deep (VZ) to superficial by unpaired two-tailed *t*-test with Welch's correction, or Mann–Whitney *U*-test, after Shapiro–Wilk normality test. *$P < 0.05$, **$P < 0.01$, ***$P < 0.001$, ****$P < 0.0001$. See also Extended Data Fig. 3a,b and Supplementary Table 3.

Weak Satb2 protein expression is first detected at E14.5, in contrast to strong Bcl11b protein now appearing in post-mitotic neurons (Fig. 2d, middle panels). Satb2 protein expression is robust only by E16.5,

concordant with an 8.2-fold increase in mass spectrometry signal and 1.4-fold upregulation of *Satb2* translation efficiency described above (Fig. 2d, bottom panels). *Satb2* mRNA and protein are broadly

expressed by E16.5. However, neurons that have migrated to their ultimate position in upper layers almost exclusively express Satb2 rather than Bcl11b protein, in contrast to regions like the intermediate zone where neurons continue to migrate.

Taken together, *Satb2* mRNA and protein expression are divergent in developmental time and space. This divergence includes broad, early *Satb2* mRNA expression in multipotent progenitors with Satb2 protein ultimately restricted to upper-layer post-mitotic neurons later in development. Whereas the distribution and colocalization of mRNA for *Bcl11b* and *Satb2* neuronal programs remains broad and overlapping, corresponding protein expression is more exclusive, with the intermediate zone a transitory region where neuronal fates still lack distinction.

### Selective Satb2 protein expression after broad transcription

Given the unexpected finding of *Satb2* mRNA in early-born neural stem cells, we next sought to monitor transcriptional activation of the *Satb2* locus. We used a fate-mapping approach with the *Satb2*[Cre/+] mouse line[36]. A Cre expression cassette is located in place of exon 2 at the *Satb2* locus, for timed in utero electroporation (IUE) of Cre-inducible reporters like *loxP-STOP-loxP-tdTomato* to clonally label cells with tdTomato that have a history of *Satb2* transcription (*Satb2*[tdTom]) (Fig. 3a). Co-electroporation with an eGFP plasmid serves as a generic label for all transfected cells.

Remarkably, we detected *Satb2*[tdTom] cells in the ventricular zone as early as E12.5 forming clusters resembling clones or undergoing mitotic divisions (Fig. 3b,c), and expressing neural progenitor markers like Pax6 (apical progenitors) or Tbr2 (intermediate progenitors) (Fig. 3c). *Satb2* transcription was observed for progenitors in the neocortex, but not in the adjacent ganglionic eminence (Extended Data Fig. 3c). Thus, *Satb2* transcriptional priming occurs in early-born neocortex neural stem cells.

The balance of Bcl11b vs Satb2 neuronal lineages is essential to form functional subcortical vs intracortical circuits, respectively. Satb2 directly suppresses the *Bcl11b* genomic enhancer, and loss of Satb2 results in ectopic expression of *Bcl11b* in upper-layer neurons, leading to abnormal connectivity[22]. Therefore, we next investigated the expression of Bcl11b and Satb2 protein in early-born cells that transcribe *Satb2* mRNA (Fig. 3d,e and Supplementary Table 3). Among stem cells transcribing *Satb2* mRNA at E12.5, ~65% express Satb2 protein and ~35% express Bcl11b protein at E14.5. Notably, both *Satb2* and *Cre* mRNA are expressed under the control of the same *Satb2* promoter; however, we detected significantly fewer Satb2 protein-positive cells than Cre protein-positive cells, further suggesting that *Satb2* translation output is distinctly regulated (Fig. 3e). Taken together, despite unexpectedly broad and early transcription of the neuronal fate gene *Satb2*, translation of Satb2 protein restricts its expression to a late-born neuronal subtype, and maintains the balance of alternative neuronal fates.

### Translation downregulation of ribosome biogenesis

Next, we focused on genes that are translationally downregulated across neurogenesis after E12.5 (Supplementary Table 2). GO analysis highlighted structural constituents of the ribosome, predominantly ribosomal proteins, as strongly downregulated by translation (Fig. 4a). We calculated the developmental expression trajectory of all 79 ribosomal proteins in the large and small subunits by RNA-seq, Ribo-seq, mass spectrometry and translation efficiency (Fig. 4b). Results showed that downregulation of nearly all ribosomal proteins at the Ribo-seq and mass spectrometry levels occurs acutely at E15.5, in advance of changes measured by RNA-seq, reflecting translation downregulation until mid-neurogenesis. Decreasing ribosome levels by downregulation of ribosomal protein translation likely represents the coordinated regulation of this specific gene family, rather than a simple translation feedback loop, as numerous genes in other families undergo translation upregulation concurrently.

To detect changing ribosome numbers subcellularly at high resolution, we performed immuno-electron microscopy analysis labeling

ribosomal protein uS7 at E12.5 and E15.5 in the neocortex (Fig. 4c,d, Extended Data Fig. 4 and Supplementary Table 3). Unlike neurons differentiating early, a striking decrease in ribosome number was observed in differentiating neurons at late stages. Ribosomes are abundant in cortical plate neurons at E12.5, but scarce in both upper-layer and lower-layer neurons of the cortical plate at E15.5. Early-born neurons emerging from the stem cell niche at E12.5 increase their ribosome number; whereas at E15.5, ribosome numbers decrease precipitously outside the stem cell niche. Thus, ribosome number is temporally enforced by translation at mid-gestation. As ribosome abundance is a powerful determinant of translation kinetics and selectivity[37,38], global shifts in translation activity may occur at mid-neurogenesis.

### Ribosome density on coding sequences is developmentally dynamic

Next, we analyzed ribosome–mRNA interactions per codon across all coding sequences. Ribosome position aligned to codons in the P-site demonstrated characteristic three-nucleotide periodicity in Ribo-seq metagene plots (Fig. 5a). We found that ribosome occupancy surrounding the start codon increases sharply at E15.5, with progressive increases per stage until P0, whereas stop codon occupancy demonstrates the opposite trend and occurs independent of start codon changes (Supplementary Table 4). We applied RiboDiPA[39], a linear modeling framework designed for positional analysis of the Ribo-seq signal, to pinpoint the ~5-fold ribosome occupancy changes to the four-codon bin surrounding the start and stop (Fig. 5b).

Increased ribosome occupancy of the first four codons over time could represent a narrowing bottleneck in the transition from initiation to elongation, or signify increasingly robust initiation of target mRNAs. We correlated fold changes in start codon occupancy with translation efficiency and found an inverse relationship, suggesting that early elongation events progressively slow over time for a large cohort of proteins (Fig. 5c). Thus, as ribosome levels decline at E15.5 to P0, translation at the 5' end of coding sequences occurs more slowly.

Then, we investigated distinct positions where variations in ribosome density take place[40] (Fig. 5d; see Methods). A narrow region consistent with the ribosomal A-site accounts for most of the codon-specific variation in ribosome occupancy. Variation in A-site occupancy was most pronounced at E12.5–E14, with an acute decrease at E15.5–E17, and low variation by P0. Analysis of ribosome dwell time per codon, a measure of the codon-specific speed of translation[41], demonstrated an early developmental 'fast' or 'slow' bimodal distribution of codon occupancy in the A-site (Fig. 5e, Extended Data Fig. 5a,b and Supplementary Table 4). At E15.5, codon occupancy begins to equalize, progressively reaching a unimodal distribution by P0. Furthermore, ribosome density occupying A-site codons negatively correlates with P-site density in the embryonic period, but no correlation was measured after birth at P0 (Fig. 5f). Thus, the A-site codon in particular influences ribosome density, which is a barrier most pronounced early in neurogenesis when ribosome levels are highest, and less pronounced after mid-neurogenesis when ribosome levels decline.

Varying ribosome occupancy of a codon might be attributed to the availability of its corresponding tRNA. Occupancy is strongly correlated with tRNA abundance in yeast[42–44], but is less correlated in mammalian systems[41,45]. We measured the levels of 151 tRNA isodecoders by qPCR array at each stage (Extended Data Fig. 6 and Supplementary Table 4) to determine if tRNA abundance is responsible for driving ribosome occupancy differences in the developing neocortex. Usage-corrected tRNA abundance (availability)[41] and codon optimality, the non-uniform decoding rate between synonymous codons[46], failed to show any correlation with ribosome dwell time at the A-site (Extended Data Fig. 5c,d).

However, we found that the amino acid coded for is a strong determinant of ribosome occupancy of A-site codons, with synonymous codons showing similar occupancy (Fig. 5g and Supplementary

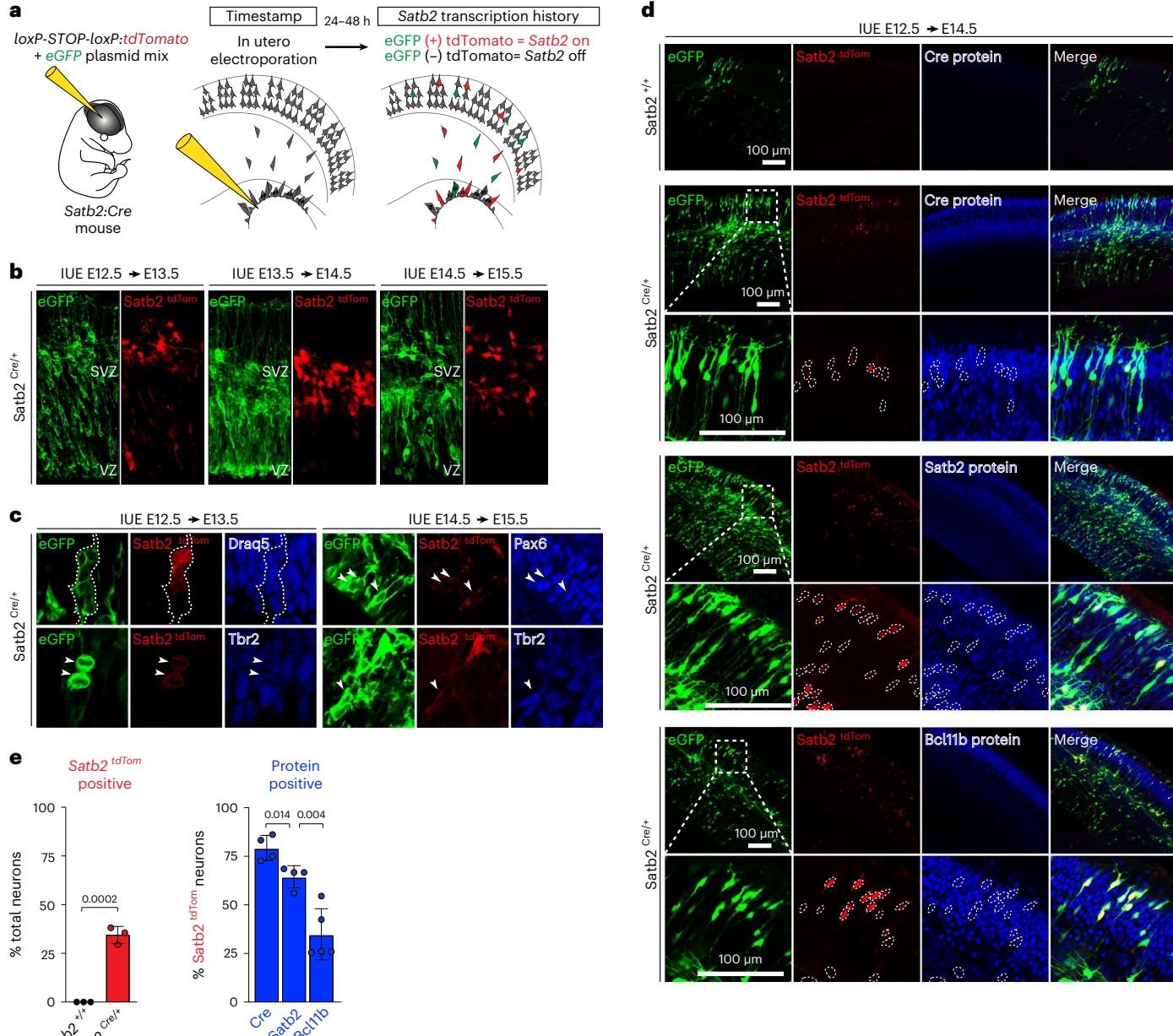

**Fig. 3 | Satb2 transcription is broad across neuronal lineages with more restricted translation. a**, Schematic of the experimental approach. **b**, *Satb2* transcription activity visualized by Cre-driven (*Satb2^Cre/+^*) tdTomato expression, with reporter IUE at E12.5, E13.5 or E14.5 and imaged after 24 h. Co-electroporation of an eGFP plasmid labels all transfected cells. SVZ, subventricular zone; VZ, ventricular zone. **c**, *Satb2^tdTom^* co-immunolabeling with Pax6 (apical progenitors), Tbr2 (intermediate progenitors) and Draq5 (nuclei). In **b** and **c**, at least three independently electroporated animals were imaged. **d**, *Satb2^tdTom^* expression at E12.5–E14.5, with co-immunolabeling for neuronal fate determinant proteins Satb2 and Bcl11b, among all electroporated cells (eGFP). Negative control is the absence of *Cre* (*Satb2^+/+^*). **e**, Quantification of **d** for the number of total neurons transcribing *Satb2* mRNA (*Satb2^tdTom^*, left), and the number of *Satb2^tdTom^* neurons synthesizing Cre, Satb2 and Bcl11b proteins (right). Mean ± s.d. is shown; *n* = 3 independently electroporated brains quantified for tdTomato, 4 for Cre, 4 for Satb2, and 5 for Bcl11b protein. Two-tailed *t*-test, *P* < 0.05 as shown. See also Extended Data Fig. 3c and Supplementary Table 3.

Table 4). Codons for acidic amino acids are among those with the highest occupancy, suggesting that they represent a kinetic barrier in early development translation[43,47]. E12.5–E14 accounts for the extremes of A-site differences between amino acids and among synonymous codons, with a progressive, chronologic trend toward equalized occupancy by P0. Notably, some amino acids like leucine and isoleucine are coded for by both 'fast' and 'slow' synonymous codons, particularly apparent early in development, such as the fast TTA-Leu and slow CTG-Leu. Neither codon optimality (Extended Data Fig. 5d) nor codon rarity would account for such dwell-time differences, as TTA-Leu is a relatively rare codon[48] with a short dwell time, whereas CTG-Leu is more common with a long dwell time.

Taken together, 'fast' and 'slow' amino acids in the ribosome A-site characterize early neurogenesis when ribosome levels are transiently abundant, whereas ribosome accumulation at the start codon occurs late in neurogenesis when ribosome levels decline (Fig. 6a). Ribosome biogenesis and translation dynamics coincide with the chronology of neuronal fate transitions at mid-neurogenesis.

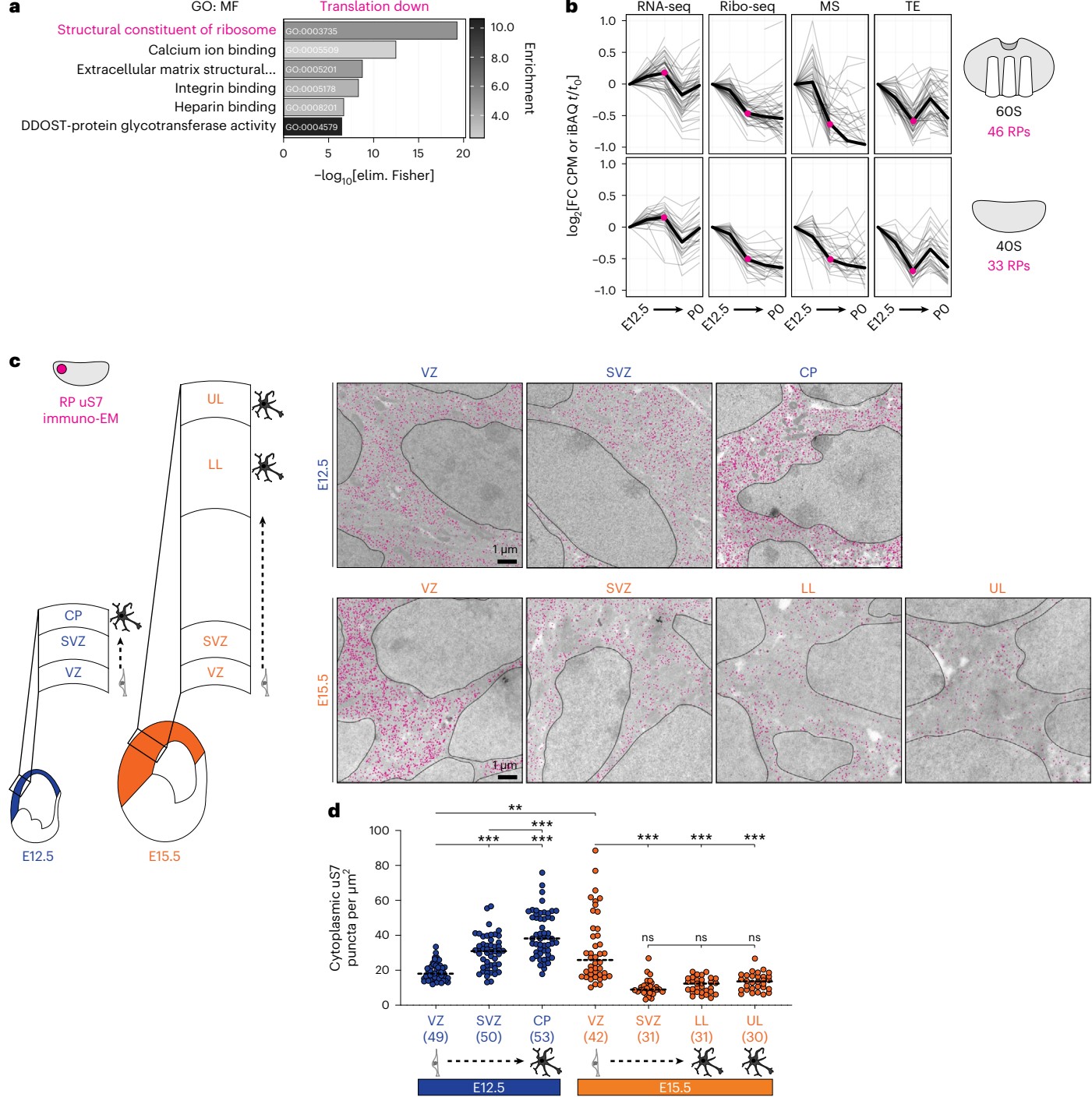

**Fig. 4 | Translation downregulation decreases ribosome levels acutely at mid-neurogenesis. a**, GO (molecular function) analysis of translationally downregulated (TE down) mRNAs. **b**, The expression trajectories (gray) of all 79 ribosomal protein coding mRNAs in the large (*Rpl*) and small (*Rps*) subunits from E12.5 ($t_0$) to subsequent stages (*t*), measured by RNA-seq, Ribo-seq, mass spectrometry and calculated translation efficiency. Median trajectories are shown in black. **c**,**d**, Immuno-electron microscopy (immuno-EM) labeling ribosomal protein uS7 (magenta) in the E12.5 and E15.5 neocortex neural stem cells and neurons (**c**), with quantification for ribosomes per cytoplasmic area (**d**); *n* = electron microscopy images, captured from 2 independent brains, 2 sections per brain at each developmental stage. Mean is shown (line), two-way Welch's ANOVA and Dunnett's post hoc test, **\**P* < 0.01, ***\**P* < 0.001. Neural stem cells are located in the ventricular zone (VZ) and subventricular zone (SVZ); post-mitotic neurons are located in the cortical plate (CP), lower layers (LL) and upper layers (UL). Nuclei are outlined. See also Extended Data Fig. 4 and Supplementary Table 3.

**Ribosome biogenesis is synced to neuronal lineage chronology**

The overwhelming influence that changes in ribosome number can have on global protein synthesis kinetics and mRNA-specific translation is strongly supported by theoretical and experimental data[37,38]. However, a mechanism regulating ribosome biogenesis during neocortex neurogenesis is unknown. We analyzed mRNAs for sequence motifs in their untranslated regions (UTRs), which are powerful regulators of neuronal translation by RNA-binding proteins[2,17,49] and initiation factors[50–52]. Distinct motifs are enriched in the 5′ UTRs and 3′ UTRs of mRNAs with increasing or decreasing translation efficiency

(Fig. 6b). Translation downregulation motifs were detected only in 5′ UTRs and are enriched for terminal oligopyrimidine (5′ TOP) sequences. In translation-upregulated mRNAs by contrast, 5′ GC-rich sequences and/or 3′ Pumilio-binding motifs are prevalent. 5′ TOP sequences are a particular feature of ribosomal proteins coding mRNAs[53], and lead to their concerted translation down-regulation by the major upstream regulator eIF4EBP1 (Fig. 6c)[24–26]. As we found that ribosome levels are controlled by a timed decrease in ribosomal protein translation, we then focused on how eIF4EBP1 activity coincides with translation regulation during neocortex development.

Western blot analysis of neocortex lysates demonstrated that eIF4EBP1 expression is high at early stages until E15.5, followed by a sharp decrease at E17, and moderate recovery at P0 (Fig. 6d). Phosphorylation of eIF4EBP1, occurring downstream of mTOR stimulation, controls eIF4EBP1 activity by triggering its dissociation from initiating ribosomes[54], which disinhibits 5′ TOP translation[24,25] (Fig. 6c). We found that eIF4EBP1 phosphorylation is most abundant at the earliest stages of neurogenesis, and declines sharply at E15.5 in advance of the decrease in eIF4EBP1 levels overall at E17 (Fig. 6d). Thus, ribosomal protein translation may remain high at E12.5 despite high levels of eIF4EBP1 due to its phosphorylation-driven dissociation from initiating complexes. Furthermore, the relative abundance of unphosphorylated eIF4EBP1 at E15.5 would permit a sharp decrease in ribosomal protein translation at this stage (Figs. 4b and 6d, bottom).

Next, we assessed cellular eIF4EBP1 expression in the developing neocortex by IHC (Fig. 6e). Robust eIF4EBP1 expression was observed in neural stem cells at E12.5–E15.5 in the ventricular zone, with lower expression in cortical plate neurons. From E17 to P0, eIF4EBP1 levels decrease globally. Phosphorylation of eIF4EBP1 occurs heavily in mitotic figures on the ventricular surface, and is enriched throughout the ventricular zone and early cortical plate from E12.5 to E14 (Fig. 6f). At E15.5, phosphorylation declines particularly in the cortical plate, with low levels throughout the neocortex from E17 to P0. These data suggest that the decrease in ribosome number observed in differentiating cortical plate neurons at E15.5 (Fig. 4c,d) may result from a decrease in eIF4EBP1 phosphorylation particularly at this time and in this location. Interestingly, eIF4EBP1 phosphorylation is minimal in postnatal glial lineage progenitors, in contrast to prenatal neurogenesis. Thus, eIF4EBP1 may play a role in the mid-neurogenic period, during a down-regulation of ribosomal protein translation, which coincides with an increase in Satb2 translation.

We reasoned that enforcing low eIF4EBP1 levels would mimic the earliest progenitor state when eIF4EBP1 is strongly inactivated by phosphorylation, which could influence production of the later-born Satb2 neuronal lineage. We performed IUE of an *eIF4EBP1* short hairpin RNA (shRNA) in ventricular zone progenitors at E13.5, followed by IHC assessment of Satb2 protein expression at E15.5 (Fig. 6g). eIF4EBP1 knockdown in early progenitors leads to a decrease in the fraction of Satb2 protein-expressing neurons at E15.5 compared with scrambled control, and arrests neuronal entry into the cortical plate (Fig. 6g,h and Supplementary Table 3). These data indicate that eIF4EBP1 affects neuronal fate and migration during a critical window for translation efficiency, ribosome biogenesis and core translation dynamics at mid-gestation (Fig. 6i).

## Modeling the translatome of neocortex neurogenesis

Next, we pursued a more comprehensive bioinformatic analysis of the transcriptome-to-proteome transition, in which deviations between mRNA and protein may represent dynamic cellular transitions[1]. Hierarchical clustering of mRNA and protein expression trajectories after E12.5 per gene divided the proteome into 13 broad clusters (Fig. 7a, Extended Data Fig. 7a and Supplementary Table 5). Clusters represented concordant and divergent trajectories between mRNA and protein, with E15.5 a common inflection point of divergent regulation. Although genes with changing translation efficiency are found in all clusters, they are enriched in clusters that demonstrate highly divergent mRNA and protein expression. Furthermore, several essential neural stem cell and differentiation markers segregate into distinct clusters such as *Nes* in cluster J and *Satb2* in cluster M. Reinforcing the biological significance of different mRNA and protein trajectories, GO analysis demonstrated that many non-overlapping, distinct pathways are enriched in different clusters, such as neuron differentiation processes enriched in cluster G (Fig. 7b and Supplementary Table 5).

The relationship between Ribo-seq density and steady-state protein levels is complicated by the fact that protein half-lives are relatively long[55], and reflect the cumulative effects of synthesis and degradation over time. In contrast, Ribo-seq reflects synthesis at a given time point. Deviations between protein and Ribo-seq are expected whenever protein levels have not yet reached equilibrium with synthesis, making linear comparison of protein concentrations and Ribo-seq densities difficult to interpret. We therefore made use of a kinetic, time-continuous model of protein translation similar to ref. [56] (see Methods).

We classified proteins into one of five categories (Fig. 7c): (1) linear, in which protein levels are in near-equilibrium with Ribo-seq measured synthesis; (2) production, consistent with a non-equilibrium protein trajectory; (3) MSdev, in which protein trajectories diverge from their Ribo-seq trajectories; (4) stationary, in which protein levels show little change; and (5) degradation, for which protein degradation alone fits the data. By using the approximation of a single constant relating RPF density and synthesis rate, we estimated half-lives for all genes, which show a strong correlation to experimentally determined degradation rates in NIH 3T3 cells[57] (Extended Data Fig. 7b). For example, our predicted MSdev category proteins are more likely to demonstrate non-exponential decay kinetics during their lifetime (Extended Data Fig. 7c).

Genes in the five modeled categories show distinct GO term enrichment, such as a linear relationship between the translation and abundance of ribosome components, or the non-linear (production) relationship for chromatin-associated proteins, including Satb2 (Fig. 7d and Supplementary Table 5). Interestingly, G-protein-coupled receptors and DNA replication genes are enriched in the MSdev category, suggesting that their expression patterns are highly multifaceted. In contrast, transmembrane transporter protein levels are highly stable, buffering upstream transcription or translation changes. Thus, our modeling highlights how multiple layers of post-transcriptional regulation affect distinct gene families during the time course of neuronal differentiation.

## Discussion

By mapping the quantitative landscape of the transcriptome-to-proteome transition in the developing neocortex, we find that protein

---

**Fig. 5 | Ribosome density at the start codon and in the coding sequence shifts sharply at mid-neurogenesis. a**, Ribosome occupancy metagene plot including all mRNAs (top) surrounding the start (left) and stop (right) codons at five stages. Separation of mRNAs by changing or unchanged start codon occupancy (bottom). **b**, Position-specific fold changes in ribosome P-site counts surrounding the start and stop codons. **c**, Start (left) and stop (right) codon occupancy vs translation efficiency fold change per gene. Center is the maximum likelihood slope; ribbon is the 95% confidence interval. **d**, Between- codon variance in ribosome occupancy of A-sites, P-sites and E-sites at each stage. Calculation with both 29 nt (top) and 30 nt (bottom) RPF fragments shown. **e**, Distribution of per-codon A-site and P-site occupancy at each stage. **f**, Correlation between A-site and P-site occupancy per codon. Center is the maximum likelihood slope; ribbon is the 95% confidence interval. **g**, Ribosome A-site occupancy for each amino acid with corresponding synonymous codons at each stage. See also Extended Data Figs. 5 and 6 and Supplementary Table 4.

synthesis is a dynamic and widespread layer of timed gene expression regulation affecting neuronal specification at mid-gestation. We find widespread deviations in the trajectory of mRNA and protein

expression along with changes in translation for ~18% of the transcriptome, with a transient peak at mid-neurogenesis. We interrogate the protein families most enriched among translation upregulated and

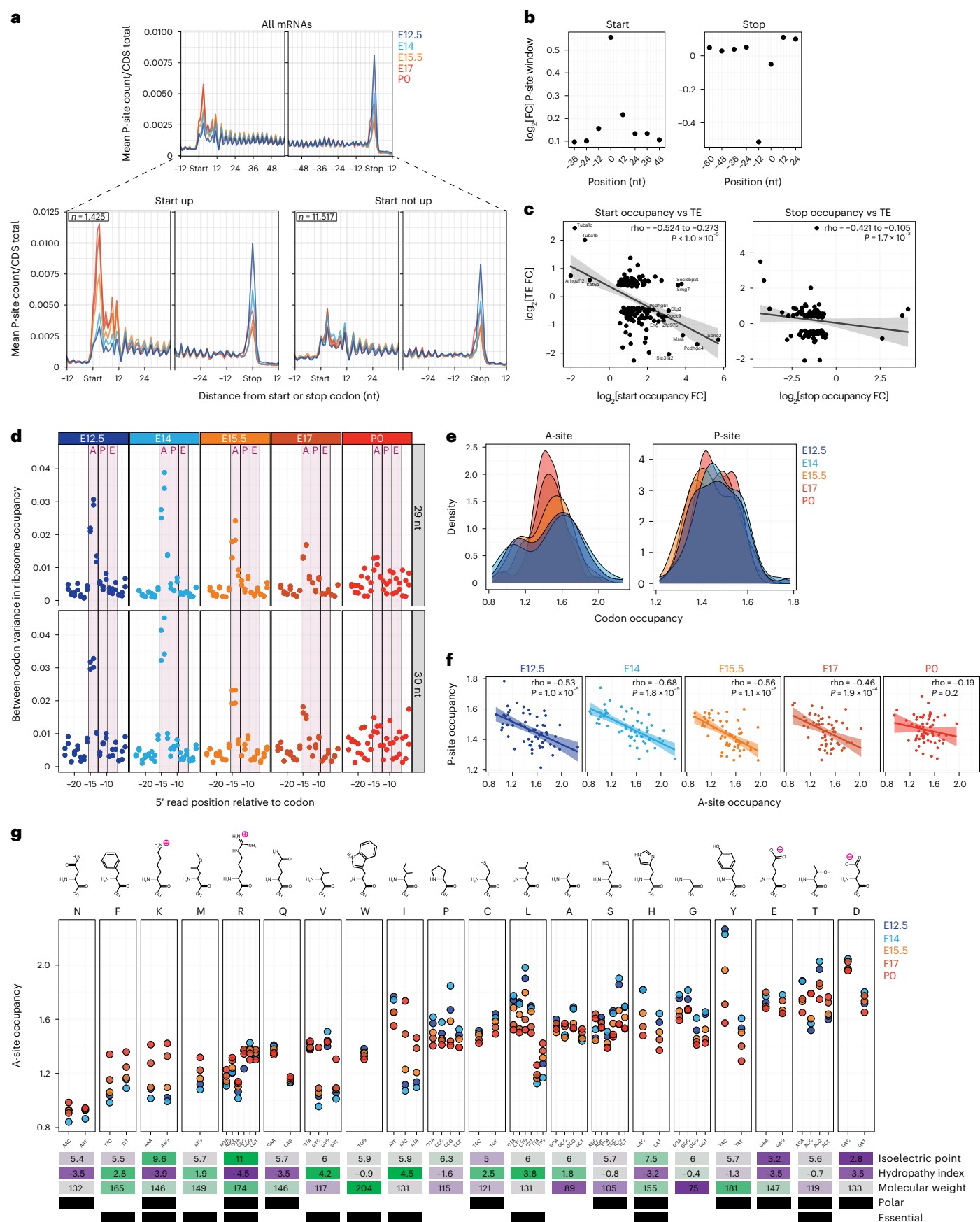

downregulated genes. Translation upregulation particularly affects chromatin-binding proteins like Satb2, which are essential drivers of neuronal subtype specification. Translation downregulation targets the translation machinery itself, with an acute decline in ribosome biogenesis at mid-neurogenesis. The transition from relative ribosome abundance to scarcity is accompanied by a chronological shift in translation processivity at the start codon and A-site amino acid during peptide elongation. Activity of eIF4EBP1, the major upstream suppressor of ribosomal protein translation, shifts in tandem with ribosome biogenesis, and affects Satb2 neuronal fate specification. Finally, our modeling highlights the effect of translation in a multilayered program of neurodevelopmental gene expression.

In more evolutionarily ancient systems like that of *Caenorhabditis elegans*, and to an extent in the mammalian spinal cord, combinatorial expression of *Homeobox* cluster transcription factors sharply demarcates distinct neuronal populations[58,59]. In the mammalian neocortex, a transcription factor code accounting for the immense cellular diversity has been elusive, and neural stem cells and differentiated neurons harbor a pool of mRNAs inclusive of diverse neuronal fates[4,5,7–9]. We propose that a broad transcriptome[60] is filtered at the protein level for tightly timed, rapidly scalable and spatially targeted gene expression to assemble highly evolved neuronal circuits. Per gene per hour, translation is faster and more scalable than transcription by orders of magnitude[55], and neuronal specification transitions occur in very narrow developmental windows[2,3]. The availability of a diverse mRNA repertoire, including both *Bcl11b* and *Satb2* fates, can be rapidly and selectively amplified by translation to specify Bcl11b or Satb2 protein exclusive neurons, or Bcl11b–Satb2 double-positive neurons[6].

We find translational downregulation of ribosome biogenesis at mid-neurogenesis. Control of ribosome number has a dominant influence on global protein synthesis kinetics and mRNA-specific translation, and can lead to 'ribosomopathies' in disease states[37,38]. With a shifting landscape of eIF4EBP1 activity and ribosome abundance, our study joins an evolving body of work on RNA-binding proteins and ribosome cofactors that modulate protein synthesis in the developing brain[2,5,14–19,49–52]. eIF4EBP1 is a master regulator of ribosome levels by suppressing ribosomal protein synthesis[24,25], playing a particularly dynamic role in stem cells downstream of mTOR[26], where we find that it affects the fate and migration of a neocortex neuronal lineage prenatally. A timed mechanism to finely tune ribosome levels may impose essential control on how and when proteins are synthesized prior to[61] and during neuronal fate decisions.

We measure a timed, progressive developmental shift in ribosome density surrounding start and stop codons. Although it has been proposed that the '5′ ramp' represents 'slow' synonymous codon choice to prevent ribosome collisions downstream[62], we find an increase in start codon density despite the generally decreasing effect of codon choice. Increasing density at the 5′ coding sequence may reflect a shift from ribosome-abundant, elongation-limited translation to ribosome-scarce, initiation-limited translation[63] as barriers to early amino-terminal peptide elongation[64] become prominent. We do not observe increasing start codon density only for genes with high translation efficiency, or correlation

with neurite-localized translation (Extended Data Fig. 8a,b). We therefore favor the hypothesis that ribosome occupancy at the beginning of ORFs becomes progressively rate limiting for codon-independent reasons, such as scarcity of ribosome machinery later in development.

We also find that the A-site amino acid is rate limiting during early neurogenesis in particular. Electrostatics[43,47], amino acid availability and/or tRNA aminoacylation might thus play a prominent role in early brain development. Our study demonstrates differences in codon-specific ribosome density over developmental time. Our findings are concordant with tRNA levels not representing a limiting resource for translation in mammals[41,45,65], in contrast to yeast[42–44]. However, these findings do not rule out individual cases in which a tRNA may influence ribosome stalling, as reported for one nervous-system-specific tRNA postnatally[66]. We measure the total tRNA pool with a protocol that does not address tRNA charging, which is a limitation of our study.

The main limitation of our study is that parallel time course measurements of the transcriptome, tRNA pool, translatome and proteome occur in brain tissue of mixed cell types rather than single cells. In addition to scRNA-seq, tremendous advances in single-cell Ribo-seq were recently published[67]. Although improvements to the depth of single-cell proteomics are still underway[68,69], the input requirements for tRNA measurement remain a major obstacle. At the expense of cellular resolution, we opted to perform a comprehensive analysis that enables modeling of mRNA translation in developing brain tissue. Notably, although our study is well designed to measure changes in protein synthesis, we do not measure protein degradation directly. The unexpectedly large number of MSdev proteins identified in our model indicates that post-translational mechanisms such as degradation[70–73] may also have a major effect. Despite these limitations, we validated two important findings at the cell-type-specific level in situ: mRNA–protein uncoupling of Satb2, and coordinated downregulation of ribosome biogenesis. We anticipate that our bulk tissue measurements can be leveraged in tandem with single-cell data[74–76].

These data open new avenues for inquiry, such as the sequence determinants of translational control in the brain. For example, from E12.5 to E15.5, ribosome density in the 5′ UTR decreases for mRNAs with downregulated translation efficiency and increases for mRNAs with upregulated translation efficiency (Extended Data Fig. 8c). Ribosome occupancy in the 5′ UTR may be indicative of upstream ORFs and/or functional mRNA secondary structures. Indeed, we find that potential G-quadruplex-forming sequences are enriched in mRNAs with upregulated translation efficiency (Extended Data Fig. 8d). Notably, although above we use the term 'translation efficiency' broadly adopted by the field, Ribo-seq data are more accurately described as 'ribosome density' on mRNA that may represent a wide range of phenomena from ribosome stalling to robust translation. High ribosome density may reflect stalled polysomes poised for translation activation in response to synaptic activity, as previously described[77].

Taken together, our data suggest a model of developmental gene expression in which the global activity and transcript specificity of translation shift during a key window of neurogenesis

**Fig. 6 | eIF4EBP1 regulation coincides with ribosome abundance and controls neuronal Satb2 fate in vivo. a**, Model of early vs late neurogenesis ribosome levels and per-codon changes in ribosome occupancy. **b**, Positional weight matrix of the top two motifs ranked by *P* value in the 5′ UTR and 3′ UTR of TE up or down mRNAs. 5′ TOP motifs are highlighted (pink square). **c**, Schematic portraying eIF4EBP1 inhibition of ribosomal protein mRNA 5′ TOP sequence translation. **d**, Western blot analysis of total and phosphorylated eIF4EBP1 levels in neocortex lysates in biological duplicate (*n* = 4–6 hemispheres per lane). Concurrent trajectory of Rpl and Rps translation is shown below. **e,f**, IHC of total (**e**) and phosphorylated (**f**) eIF4EBP1 expression in neocortex coronal

sections across neurogenesis. Blood vessels (stars) are a common staining finding. **g**, shRNA knockdown of eIF4EBP1 compared with scrambled control by IUE at E13.5, followed by analysis at E15.5 with Satb2 protein immunolabeling. Co-electroporation of eGFP labels all transfected cells. Cortical plate (CP) boundary is demarcated (dotted line), zoom of yellow boxes (right). **h**, Quantification of **g**, *n* = independent electroporated brains, for the percentage of electroporated cells expressing Satb2 protein (left), and number of cells migrating into the cortical plate (right). Median (line), two-sided Mann–Whitney *U*-test, *P* < 0.05 as shown. **i**, Summary of timed translation changes and neuronal specification during neocortex development. See also Supplementary Table 3.

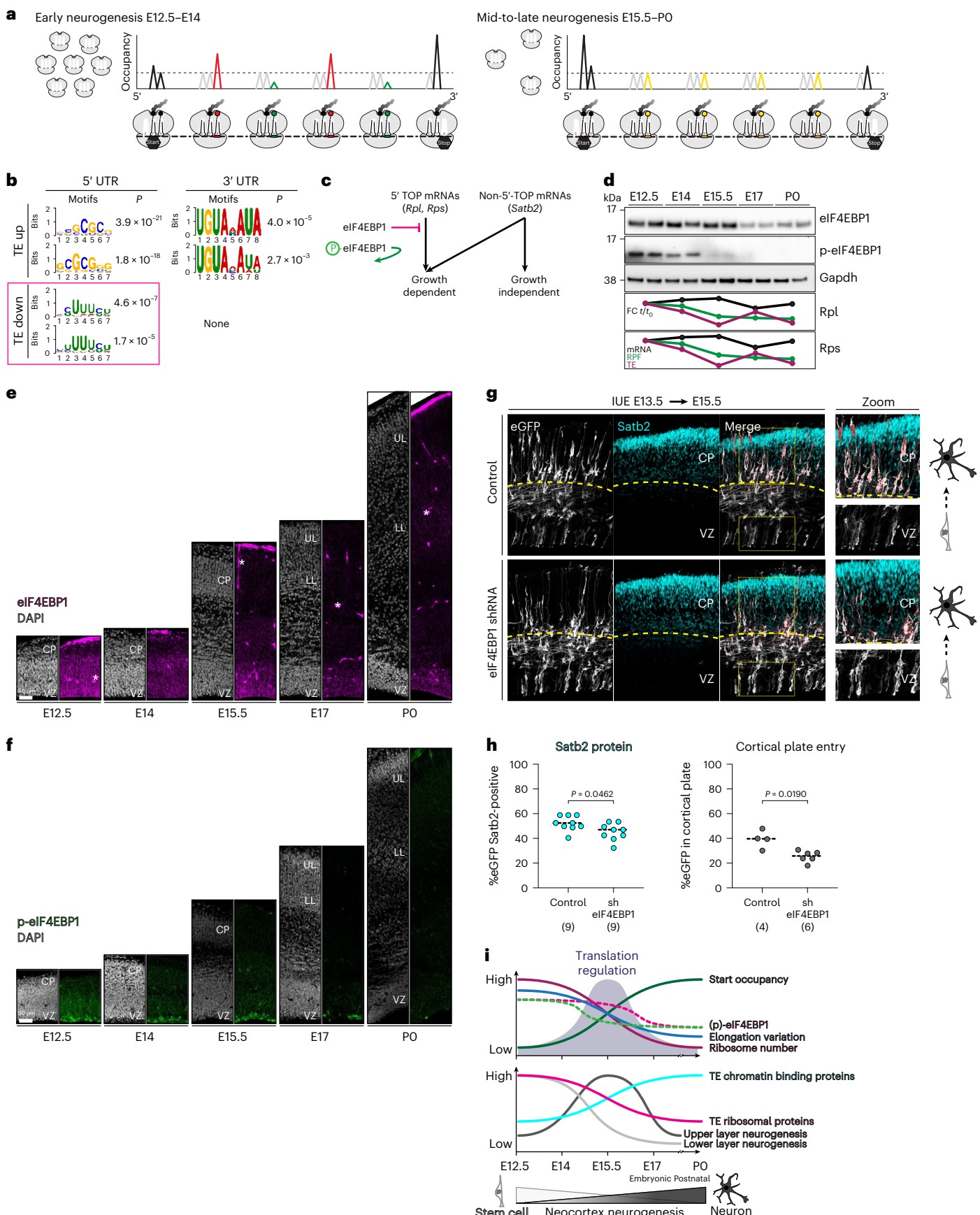

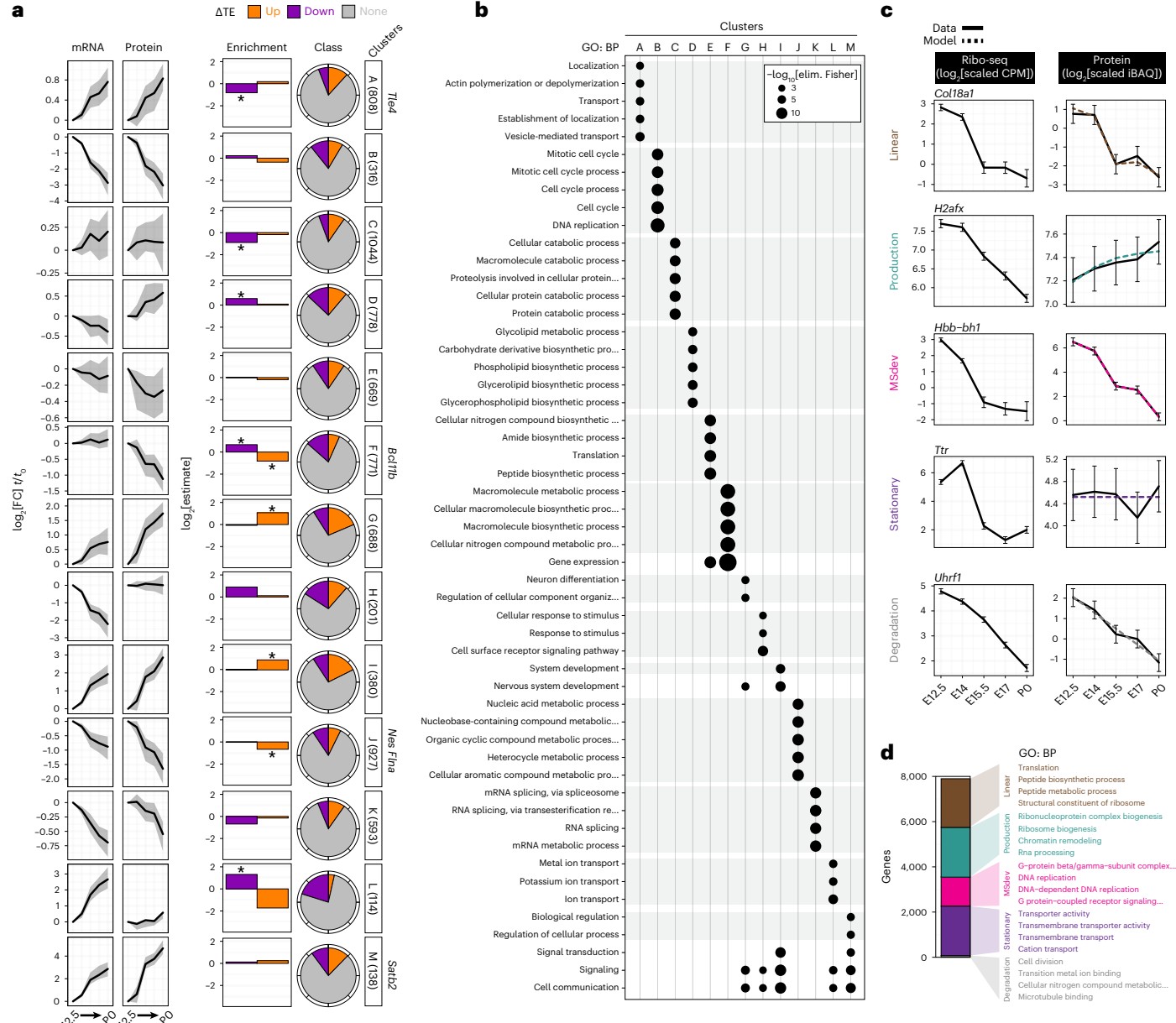

**Fig. 7 | Modeling divergent trajectories of mRNA and protein expression by translation regulation. a**, mRNA (RNA-seq) and protein (mass spectrometry) expression per gene from E12.5 ($t_0$) to subsequent stages ($t$) clustered by trajectory. The median trajectory is shown, with upper and lower quartile ribbons. Enrichment and proportion of TE up and down genes in each cluster, with significant enrichment (*$P < 0.05$). Fisher's exact test comparing TE up or down vs no change, within each class. Example neural stem cell and neuronal marker genes are indicated (right). **b**, GO (biological process, BP) enrichment for each cluster, with unique terms for a cluster outlined in gray. **c**, Modeling of non-linear relationships between Ribo-seq and mass spectrometry comparing active translation vs steady-state protein, with representative genes shown for each category. A 95% confidence interval on the model fit is shown, $n = 2$ Ribo-seq, or 3 mass spectrometry, analyses of biologically independent pooled neocortex lysates; see Main and Methods for details. **d**, Proportion of total genes in each category from **c**, with enriched GO terms per category. Fisher's exact test, $P < 0.05$. See also Extended Data Fig. 7, Supplementary Table 5 and https://shiny.mdc-berlin.de/cortexomics.

in the brain, an inflection point of translation regulation at mid-gestation (Fig. 6i), during a critical period for neurodevelopmental disorders[20,78]. Transcription of neuronal subtype-specific programs may be ultimately refined by translational control, more precisely demarcating the boundaries of neuronal circuits in the mammalian brain.

## Online content

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

[1]Berlin Institute for Medical Systems Biology, Max Delbrück Center for Molecular Medicine, Berlin, Germany. [2]Institute of Cell Biology and Neurobiology, Charité-Universitätsmedizin Berlin, corporate member of Freie Universität Berlin, Humboldt-Universität zu Berlin, and Berlin Institute of Health, Berlin, Germany. [3]Institute of Neuroscience, Lobachevsky University of Nizhny Novgorod, Nizhny Novgorod, Russian Federation. [4]Max Planck Institute for Molecular Genetics, Berlin, Germany. [5]RIKEN Center for Integrative Medical Sciences, Yokohama, Japan. [6]Max Delbrück Center for Molecular Medicine, Berlin, Germany. [7]Institute of Neuroanatomy, Charité-Universitätsmedizin Berlin, Berlin, Germany. [8]Institute of Biology, Humboldt University of Berlin, Berlin, Germany. [9]Institute of Medical Physics and Biophysics, Charité-Universitätsmedizin Berlin, Berlin, Germany. [10]Department of Computer Science, Humboldt University of Berlin, Berlin, Germany. [11]These authors contributed equally: Dermot Harnett, Mateusz C. Ambrozkiewicz [12]These authors jointly supervised this work: Victor Tarabykin, Uwe Ohler, Matthew L. Kraushar. ✉e-mail: dermot.p.harnett@gmail.com; mateusz-cyryl.ambrozkiewicz@charite. de; matthew.kraushar@molgen.mpg.de

## Methods

### Mice

Mouse (*Mus musculus*) lines were maintained in the animal facilities of the Charité University Hospital and Lobachevsky State University. All experiments were performed in compliance with the guidelines for the welfare of experimental animals approved by the State Office for Health and Social Affairs, Council in Berlin, Landesamt für Gesundheit und Soziales (LaGeSo), permissions T0267/15, G0079/11, G206/16 and G54/19, and by the Ethical Committee of the Lobachevsky State University of Nizhny Novgorod. Mice were housed in a 12 h/12 h light/dark cycle, at a consistent 18–23 °C, 40–60% humidity, with pellet food and water available ad libitum. Mice were used in the embryonic (E12.5–E17) and early postnatal (P0) period, with the stage and replicate numbers as reported for each experiment. Each sample was inclusive of both male and female sexes in each litter without distinction. Timed pregnant wild-type CD-1 mice utilized for Ribo-seq, RNA-seq, tRNA qPCR array, mass spectrometry and immuno-electron microscopy were obtained from Charles River Laboratories (protocol T0267/15). FISH and IHC experiments were performed in NMRI wild-type mice. For experiments with the tdTomato reporter, *Satb2*[Cre/+] males[22] were mated to NMRI wild-type females (protocols G0079/11, G54/19 and G206/16). *Satb2*[Cre/+] mouse genotyping was performed as described[22].

### Neocortex tissue preparation

Mouse neocortex tissue was dissected in a 4 °C room in ice-cold phosphate buffered saline, then frozen as tissue pellets in 1.5 ml tubes on dry ice, and stored at −80 °C. Frozen tissue pellets were lysed by cryogenic grinding on ice in the appropriate lysis buffer for downstream applications, and clarified by centrifugation for post-mitochondrial or post-nuclear supernatants as required (see below for the specifics for each application). Further details were previously described[18].

### Ribo-seq and RNA-seq

Each replicate for paired neocortex Ribo-seq and RNA-seq included 40 brains (80 hemispheres) at E12.5, 30 brains (60 hemispheres) at E14, 21 brains (42 hemispheres) at E15.5, 20 brains (40 hemispheres) at E17, and 17 brains (34 hemispheres) at P0, performed in biological duplicate at each stage. Neocortex tissue was lysed on ice in 20 mM HEPES, 100 mM KCl, 7.5 mM MgCl$_2$, pH 7.4, supplemented with 20 mM dithiothreitol (DTT), 0.04 mM spermine, 0.5 mM spermidine, 1x cOmplete EDTA-free protease inhibitor (Roche, 05056489001), 0.3% v/v IGEPAL CA-630 detergent (Sigma, I8896), and clarified by centrifugation at 16,100$g$ for 5 min at 4 °C with a benchtop centrifuge. Samples were then measured for A260 ODU on a NanoDrop 1000 Spectrophotometer. Two-thirds of the sample were transferred to a new tube for Ribo-seq preparation, and the remaining one-third for RNA-seq was mixed with 100 U SUPERase-In RNase inhibitor (Thermo Fisher, AM2694) and frozen at −80 °C for downstream RNA isolation.

For digestion of RPFs, Ribo-seq samples were then mixed with 60 U RNase T1 plus 96 ng RNase A per ODU, and incubated for 30 min at 25 °C, shaking at 400 r.p.m. To stop RNase activity, we then added 200 U of SUPERase-In RNase inhibitor.

A total of 5 ml of 10–50% sucrose density gradients were prepared in Beckman Coulter Ultra-Clear Tubes (344057). Base buffer consisted of 20 mM HEPES, 100 mM KCl, 10 mM MgCl$_2$, 20 mM DTT, 0.04 mM spermine, 0.5 mM spermidine, 1x cOmplete EDTA-free protease inhibitor, 20 U ml$^{-1}$ SUPERase-In RNase inhibitor, pH 7.4, prepared with 10% and 50% sucrose w/v. Overlaid 10% and 50% sucrose-buffer solutions were mixed to linearized gradients with a BioComp Gradient Master 107ip.

Digested lysates were overlaid on gradients pre-cooled to 4 °C. Gradients were centrifuged in an SW 55 rotor (Beckman Coulter) for 1 h at 4 °C and 166,000$g$, and fractionated using a BioComp Piston Gradient Fractionator and Pharmacia LKB Superfrac, with real-time A260 measurement by an LKB 2238 Uvicord SII UV detector recorded using an ADC-16 PicoLogger and associated PicoLogger software.

Fractions corresponding to digested 80S monosomes were pooled and stored at −80 °C.

RNA isolation with TRIzol LS was then performed for both RNA-seq and Ribo-seq samples, as per the manufacturer's instructions. For Ribo-seq and RNA-seq samples, downstream library preparation and sequencing were performed as described[18]. RNA-seq data were used in a recent study[18] corresponding to National Institutes of Health (NIH) Gene Expression Omnibus (GEO) entry GSE157425. Ribo-seq data in this study are deposited in the NIH GEO under accession number GSE169457.

### Mass spectrometry

Total proteome analysis from neocortex lysates at E12.5, E14, E15.5, E17 and P0, including complete lysis in RIPA buffer, and downstream processing for mass spectrometry analysis, was performed in a recent study[18] corresponding to ProteomeXchange entry PXD014841.

### tRNA qPCR array

tRNA qPCR array measurement of 151 tRNA isodecoders was performed by Arraystar for neocortex lysates at E12.5, E14, E15.5, E17 and P0 from the same total RNA isolated for RNA-seq described above (Supplementary Table 4). Data are deposited in the NIH GEO under accession number GSE169621.

### Ribo-seq and RNA-seq data processing

Raw sequence data was converted to FASTQ format using bcl2fastq (https://support.illumina.com/downloads/bcl2fastq-conversion-software-v2-20.html). Adapters (sequence TGGAATTCTCGGGTGCCAAGG) were removed from Ribo-seq reads using cutadapt (v1.18)[79], as well as sequences with a quality score less than 20 or a remaining sequence length less than 12, and after removing duplicate read sequences, 4-base-pair (bp) unique molecular identifiers were trimmed from either end of each sequence using a custom Perl script. Ribo-seq reads were then aligned to an index of common contaminants (including tRNA, rRNA and small nucleolar RNA (snoRNA) sequences) using Bowtie 2 (ref. [80]). The resulting processed read files were then aligned to coding sequences (the pc_transcripts fasta file), and separately, to the genome, from GENCODE release M12 (*M. musculus*) using STAR (v2.6.1a)[81], with the following settings: STAR–outSAMmode NoQS–outSAMattributes NH NM–seedSearchLmax 10–outFilterMultimapScoreRange 0–outFilterMultimapNmax 255–outFilterMismatchNmax 1–outFilterIntronMotifs RemoveNoncanonical. RNA-seq and Ribo-seq libraries achieved high coverage, with a median of 33 million and 12 million reads mapped to protein coding transcripts, respectively. For quality control, downstream analysis focused on coding sequences with 32 or more Ribo-seq footprints in at least one stage as per ref. [29], which resulted in a set of 12,228 translated GENCODE transcripts (Supplementary Table 1).

Linear fold changes were calculated for RNA-seq and Ribo-seq using limma[82], for translation efficiency using xtail (v1.1.5)[83], and for mass spectrometry using proDA (https://github.com/const-ae/proDA) (Supplementary Table 2).

As ribosomes with their A-site over a given position will produce a distribution of read lengths mapping to nearby positions, A-site or P-site alignment represents a crucial step in the processing of Ribo-seq datasets. Frequently, algorithms for A-site alignment rely either explicitly[42,84] or implicitly[85] on the presence of large peaks at the start and/or stop codons, the known location of which provides a 'true positive' that can be used to choose P-site offsets for each read length. We found that such methods gave inconsistent results in our data, with optimal P-sites being chosen at biochemically implausible values (for example, at 0 bp from the read 5′ end). This is likely due to (1) the variable occupancy of the start or stop peak in our data, and (2) the presence of cut-site bias in our data due to the necessity of RNase T1 and RNase A digestion. Calculating RUST scores and 'metacodon'[40] plots of RPF

5′ end occurrence showed that the most variation between different codons and time points (other than cut-site bias itself at RPF termini) was nonetheless limited to a narrow region a consistent distance from the codon, for each read length. Plotting Kullback–Leibler divergence between observed and expected RUST scores at different distances from the read 5′ end and measuring the between-codon variance at each position revealed that it aligned with an offset of approximately 14–15 nucleotides (consistent with the A-site position) for reads of length between 25 and 31, so we chose these for further analysis of ribosome dwell time (Supplementary Table 4). We also observed an adjacent region of lesser variability 3 bp toward the RPF 5′ end, consistent with a non-zero but significantly less influence of the P-site codon (Supplementary Table 4).

The program DeepShape-prime[86], modified to accept our chosen P-site offsets instead of hardcoded ones, was then used to derive isoform-specific abundance measurements for each protein coding transcript.

In parallel to the above, isoform-level quantification of the RNA-seq was carried out using Salmon (v0.14.1)[87], with an index built from coding M12 sequences, and the following settings: salmon quant -l SR –seqBias –validateMappings.

A Snakemake[88] file automating the above workflow is available at https://github.com/ohlerlab/cortexomics.

We then converted the output of DeepShape-prime to Salmon format to combine both outputs, using the ORF length as effective length. The R package tximport[89] was used to derive length-corrected gene-level counts and isoform-level counts, as well as TPMs for both datasets. The voom package[82] was used for variance stabilization and linear modeling of this data to derive confidence intervals for transcriptional and translational change, both relative to E12.5, and stepwise between each stage. The xtail (v1.1.5) package[83], which is specifically geared toward estimation of translation efficiency (that is, the ratio of Ribo-seq density to RNA-seq density) change in the presence of transcriptional change, was used to detect changing translation efficiency. Numbers for translation efficiency change quoted in the text refer to xtail's differential translation efficiency calls, with stepwise fold changes shown in Fig. 1, and translation efficiency changing genes being elsewhere defined relative to E12.5 (Supplementary Table 2). Of note, six transcripts demonstrate translation efficiency changes both up and down during the five-stage time course.

For metagene plots, a 'best transcript' (the transcript with the highest median Ribo-seq coverage across all samples) was selected for each gene. These transcripts were further limited to those with a length of 192 or greater. Each of these transcripts was also analyzed using the RiboDiPA[39] package, which looks for position-specific differences in Ribo-seq occupancy between conditions. As metacodon plots indicated that changes at the start and stop codon were limited to a distinct region three or four codons from the start and stop, we divided each coding sequence into 15 bins, with seven bins of four codons each centering on the start and stop, and a final 'mid' bin of variable size encompassing the rest of the ORF (ORFs too short to accommodate this were excluded). We then plotted bin-level $\log_2$ fold changes for each gene with significant $q$ value of using the AUG/stop changing bins.

Fold changes were binarized into 'significant' (absolute fold change greater or less than 1.25, adjusted $P$ value < 0.05) and 'non-significant' for plotting upregulated and downregulated genes, respectively, and GO term analysis, referred to as dTE and non-dTE in the case of translation efficiency fold change. The R package topGO was used for GO term analyses of translation efficiency change and positional Ribo-seq change.

A list of ribosomal proteins for the mouse large and small subunits was curated from UniProt.

## tRNA abundance and codon dwell-time analysis
tRNA abundance was calculated from Arraystar Ct values by the $-\Delta Ct$ value for each tRNA compared with the mean of 5S and 18S rRNA levels

in each sample (Supplementary Table 4). Abundance per codon was calculated by taking the mean of each replicate and summing values for all relevant isodecoders. Availability[41] was calculated as the residual from a simple linear model regressing codon usage against abundance, in which codon usage was defined as the occurrence of that codon in the M12 coding transcriptome, weighted by the relevant TPM of each transcript in that sample. We attempted weighting by wobble base pairs as in ref. [46] and found this did not affect the conclusions.

We followed the approach of ref. [40] and used RUST values as a robust estimator of codon-specific dwell times. A-site occupancy was defined as the RUST value for that codon at the point of maximum variation (14 bp or 15 bp from the 5′ end) with P-site occupancy defined as the RUST value 3 bp closer to the 5′ from there (Supplementary Table 4)

Relationships between codon dwell time, tRNA abundance or availability, and amino acid identity were investigated using the R function lm. The dataset used consisted of 269 (that is, one per quantified codon, per sample) with terms for the stage of the sample, the amino acid coded for, and the abundance (or availability) of the encoding tRNAs.

The largest explanatory variable was amino acid coded for, which also showed a significant interaction with sample stage, indicating that the amino acid coded for explained ~34% of the variance in dwell time between codons. This term also showed a significant interaction with sample stage, indicating that the amino-acid-specific factors determining dwell time may vary over development (for example, owing to the availability of amino acids changing). Within a sample or across all samples, there was no association between tRNA abundance and dwell time, even after correcting for the effect of amino acid coded for. However, some codons show a significant interaction between abundance and developmental stage, and because these codons were biased toward the high or low end of the abundance–dwell-time spectrum, we plotted time-relative change vs abundance for the top and bottom quartiles of abundance–dwell-time. This revealed a significant association between change in time-relative tRNA abundance and dwell time, with fastest codons showing decreasing tRNA abundance as they slowed, and the slowest codons also showing decreasing tRNA abundance.

## Mass spectrometry data processing
All raw data were analyzed and processed using MaxQuant (v1.6.0.1)[90] (Supplementary Table 1). Default settings were kept, except that 'match between runs' was turned on. Search parameters included two missed cleavage sites, cysteine carbamidomethyl fixed modification and variable modifications including methionine oxidation, protein N-terminal acetylation and deamidation of glutamine and asparagine. The peptide mass tolerance was 6 ppm, and the tandem mass spectrometry (MS/MS) tolerance was 20 ppm. A minimal peptide length of seven amino acids was required. The database search was performed using Andromeda[91] against the UniProt/Swiss-Prot mouse database (downloaded January 2019) with common serum contaminants and enzyme sequences. The false discovery rate was set to 1% at peptide spectrum match level and at protein level. Protein quantification across samples was performed using the label-free quantification (LFQ) algorithm[92]. A minimum peptide count required for LFQ protein quantification was set to two. Only proteins quantified in at least two out of the three biological replicates were considered for further analyses.

To improve the match between mass spectrometry data and sequence data, the peptides from each mass spectrometry group were matched against M12 protein sequences. Instances in which a UniProt gene identifier did not match any gene in GENCODE, but in which the associated peptide sequences matched proteins for a single GENCODE gene, were updated to match that GENCODE gene. All further analyses were carried out using gene-level proteomic data.

The R package proDA was used to calculate dropout-aware abundance estimates for each protein group, as well as fold changes and

confidence intervals relative to E12.5. For each gene, a 'best' matching protein group was defined as the one with the least missing, and highest median, signal across all samples, and was selected for further analysis.

## Analysis of variance (ANOVA)

ANOVA was carried out a manner similar to that of ref. [30]. We fit a linear model regressing measured protein levels, or protein fold changes, P, against measured Ribo-seq or RNA-seq levels R. We then performed variance decomposition using the following equation:

$$\hat{\sigma}^2_{PDT} = \hat{\sigma}^2_{all} - \left(\frac{b_{all}}{b_R}\right)^2 \hat{\sigma}^2_R - \hat{\sigma}^2_P$$

where $\hat{\sigma}^2_{all}$ represents total variance in measured protein abundance (that is, in proDA-normalized LFQ values) and is decomposed into stochastic error in protein measurement $\hat{\sigma}^2_P$ (estimated standard error of the protein abundance model fit using proDA), systematic variation in protein levels independent of R $\hat{\sigma}^2_{PDT}$, and error in R measurement, where $b_{all}$ is the linear coefficient relating Ribo-seq and RNA-seq measurements to protein abundance, $b_R$ is the measurement bias for R, and $\hat{\sigma}^2_R$ is the stochastic measurement error in R. Lacking a means of measuring $b_R$ in our data, we experimented with a range of values, including the experimentally determined value of 1.21 based on NanoString measurements by ref. [39]. We found that owing to the relatively minor stochastic error in measurements of R, our estimates of $\hat{\sigma}^2_{PDT}$ were robust to reasonable values of $b_R$ (between 0.75 and 1.5), so we elected to fix its value at 1. We then calculated variance explained as

$$\frac{\hat{\sigma}^2_{MP} - \hat{\sigma}^2_P - \hat{\sigma}^2_{PDT}}{\hat{\sigma}^2_{MP} - \hat{\sigma}^2_P}.$$

We applied this equation both within each time point, and to the fold changes between each time point. Stochastic error terms for both within-stage and between-stage values for R and P were calculated using limma and proDA, respectively. Notably, correlation between the two sequencing assays and mass spectrometry is strongly dependent on the magnitude of change at that time point, with technical noise specific to each assay non-correlated[1]. For the R implementation of the above equations, see our github repository (https://github.com/ohlerlab/cortexomics) and the file src/Figures/Figure4/2_vardecomp.R.

## Hierarchical clustering

For hierarchical clustering (Supplementary Table 5), we took fold changes in RNA-seq and mass spectrometry values relative to E12.5, for each gene, and carried out principal component analysis on the resulting $n \times 8$-dimensional matrix. We calculated Euclidean distances between genes and performed hierarchical clustering using the R function hclust and the 'ward' clustering criterion; that is, favoring the creation of large clusters rather than small clusters containing few outliers. We found that our expression data showed a smooth reduction in variance explained as the number of clusters varied, so we plotted GO term enrichment for different cluster numbers and found that clusters with similar GO term enrichments began to appear at a cluster number of 13, so we chose this as our cutoff. Meta-trajectories for each cluster were plotted using the median and upper or lower quartiles for each cluster. Enrichment of dTE genes in each cluster was calculated using Fisher's exact test (with dTE status, and inclusion in the cluster, as binary variables). GO term analysis of each cluster was carried out using topGO.

## Non-linear trajectory modeling

To model the non-linear relationship between steady-state protein levels and Ribo-seq, a measure of protein synthesis, we used an approach similar to that used in ref. [56]; our full 'production' model

represents the expression of each protein as the result of a synthesis rate, directly proportional to Ribo-seq footprint density with a proportionality constant $K_s$ (see ref. [43]) and a decay rate $K_d$, with $R_g$ the RNA level for that protein at that timepoint:

$$\frac{d(P)}{dt} = K_s R_g(t) - K_d P(t)$$

If the functional form of Ribo-seq density is modeled as a linear stepwise function, this equation has an analytic solution[56]. In practice, the parameters $K_s$ and $K_d$ will be non-identifiable depending on the trajectory shape and half-life of the protein involved; for many proteins, only their ratio, defining the equilibrium steady state, can be estimated (along with the initial value of P). In addition to the production model, we included reduced versions of our model that fixed $K_d$ at a high value (the 'linear' model) giving a linear protein–Ribo-seq relationship, fixed $K_s$ at a low value and modeled protein as controlled by degradation only (the 'degradation' model), or fixed both to leave protein levels stationary (the 'stationary' model). We further included a model allowing arbitrary deviations from the Ribo-seq trajectory (the 'MSdev' model), as many proteins showed changes in their trajectory that were not explicable by any value of $K_s$ and $K_d$. We used the Bayesian information criterion to select an optimal model for each gene, further requiring that residuals in this model be normally distributed (as per a $\chi^2$ test). To estimate half-lives, we made the simplifying assumption of a single $K_s$ value applying to all genes, allowing $\pi$-half estimates to be derived for all proteins.

Stan files detailing the above models are available on the project github, and data are in Supplementary Table 5.

## Single-cell RNA-seq data

scRNA-seq data were derived from data and scripts in ref. [4], and the accompanying web resource http://genebrowser.unige.ch/telagirdon/#query_the_atlas.

For each gene, its occurrence in neocortex cells measured by scRNA-seq is presented as a heat map arranged by chronological time of cell collection ($x$ axis) vs time since cell birth ($y$ axis), after a timed pulse with a FlashTag label in utero. These axes correspond to roughly orthogonal programs of gene expression change, with the $y$ axis describing differences between apical progenitors and differentiated neurons, and the $x$ axis describing differences between cells born at different stages of development.

## Sequence motif analysis

Motif analysis was performed with the AME program from the MEME Suite (v5.1.1) as per refs. [56,93]. We observed a systematic difference in UTR length between genes with changing translation efficiency and genes with unchanging translation efficiency. AME requires that input and control sequences are of approximately equal length distribution, so we created a sample of genes with changing translation efficiency whose length distribution matched that of the genes with unchanging translation efficiency. We ran AME with the CISBP-RNA database of RNA-binding protein motifs[94].

## 5′ UTR ribosome density and mRNA structure analysis

Ribosome density upstream of the main ORF (mORF) was quantified by counting mapped P-sites from Ribo-seq reads in the 5′ UTR. The 15-nucleotide window immediately upstream of the mORF's start codon was excluded from counting to account for blurring and avoid including counts from the mORF. P-site counts from the 5′ UTR were then normalized to RNA abundance measured from RNA-seq as TPM of the full-length transcript. The fold change in this value from the first time point E12.5 to E15.5 was calculated for all transcripts meeting the minimum expression thresholds: at least 0.1 TPM from RNA-seq and

at least 10 P-site counts from Ribo-seq. Transcripts were grouped as downregulated, unchanged and upregulated according to their differential translation efficiency relative to E12.5 as determined by xtail above. Differences in the distribution of fold changes in 5′ UTR translational activity between these groups were tested through unpaired Wilcoxon tests.

The RNA sequences of transcript 5′ UTRs were scanned for potential to fold into an intramolecular G-quadruplex using the Bioconductor package pqsfinder (v2.10.1)[95] with default scoring settings. Counts of non-overlapping potential quadruplex-forming sequences (PQS) were normalized to 5′ UTR length. Similar to our analysis of translation activity in the 5′ UTR, the distributions of PQS density in translationally downregulated, unchanged and upregulated transcripts were compared with the unpaired Wilcoxon test.

### Immuno-electron microscopy
Fixation, sectioning, immunolabeling and electron microscopy were performed as described previously[18]. E12.5 and E15.5 neocortex coronal sections were labeled with mouse anti-Rps5 (uS7; 1:1,000, Santa Cruz Biotechnology, sc-390935) followed by 2.5 nm Nanogold-conjugated secondary antibody (Nanoprobes, 2001). Imaging was performed at ×2,700 magnification on a Tecnai Spirit electron microscope. Quantification was performed in Fiji[96] with the Process > Find Maxima tool, and Measure > Area tool, followed by statistical analysis in GraphPad Prism to calculate puncta per µm$^2$ (Welch's ANOVA and Dunnett's post hoc test) (Supplementary Table 3). Primary antibody leave-out controls were prepared in parallel, and were absent of Nanogold signal.

### Expression vectors
For tdTomato reporter experiments, we used β-actin-driven expression constructs *pCAG-EGFP* and *pCAG-flox-STOP-flox-tdTomato*, as described previously[36]. A control vector with scrambled non-silencing shRNA[72] was obtained from Thermo Scientific, and the shRNA to knock-down plasmid for mouse *eIF4EBP1* was obtained from Sigma-Aldrich (TRCN0000335381).

### In utero electroporation (IUE)
Mouse embryos were subjected to IUE exactly as described previously[36,72,97]. For the experiments with the tdTomato reporter in the *Satb2*$^{Cre/+}$ line, we used an equal amount of *pCAG-GFP* and *pCAG-flox-stop-flox-tdTomato*.

### Fluorescence in situ hybridization (FISH)
In situ hybridization using RNAscope Technology to detect mRNA of *M. musculus Satb2* (413261) and *Bcl11b* (413271-C2) was performed according to the manufacturer's protocols (ACDBio, 323100). Prior to hybridization, embryonic brains at E12.5, E14.5 and E16.5 were collected in PBS, and fixed in 4% PFA/PBS prepared with DEPC for 16–20 h at 4 °C. Brains were then incubated in sucrose solutions (10%–20%–30%/PBS) until they reached osmotic equilibrium, embedded in O.C.T. Compound (Tissue-Tek) in a plastic cryoblock mold, and frozen on dry ice. Coronal sections with a thickness of 16 µm were collected using a cryostat.

### Western blot
Analysis of neocortex lysates by western blot was performed as described[18]. The following primary antibodies were used: anti-eIF4EBP1 (1:1,000, rabbit, Abcam, ab32024, RRID:AB_2097990), anti-phospho-eIF-4EBP1 Thr37/46 (1:1,000, rabbit, Cell Signaling Technology, 2855, RRID:AB_560835), and anti-Gapdh (1:1,000, mouse, Millipore, MAB374, RRID:AB_2107445). All HRP secondary antibodies were used at a dilution of 1:2,500: HRP anti-mouse heavy chain (goat, Abcam, ab97245; RRID:AB_10680049) and HRP anti-rabbit heavy chain (goat, Cell Signaling Technology, 7074S, RRID:AB_2099233).

### Histological sectioning
For histological procedures in Figs. 2, 3 and 6g, coronal brain sections were prepared on a Leica CM3050 S cryostat. Prior to cryosectioning, brains were incubated for at least 5 h with 10% sucrose in PBS, followed by incubation with 30% sucrose in PBS until the tissue reached osmotic equilibrium. Next, brains were frozen in −38 °C to −40 °C isopentane (Roth). For processing of the tissue after IUE, coronal cryosections with a thickness of 50 µm were collected in PBS/0.01% sodium azide solution. For in situ hybridization and the mRNA or protein colocalization experiments, 16 µm sections were collected. For IHC in Fig. 6e,f, coronal brain sections were prepared as described[18].

### Immunohistochemistry (IHC)
For Figs. 2, 3 and 6g, fixed brain sections were washed with PBS three times at room temperature (20–25 °C) prior to the procedure to remove the sucrose and freezing compound residue. The sections were then incubated with blocking solution (5% goat serum, 0.5% (v/v) Triton X-100, PBS) for 1 h at room temperature, then with the primary antibody and 4,6-diamidino-2-phenylindole (DAPI) diluted in blocking buffer for 16–20 h at 4 °C, washed in PBS three times for 30 min and incubated with secondary antibody diluted in the blocking buffer for up to 4 h at room temperature. Next, sections were incubated with PBS for 30 min three times and mounted with a cover glass (Menzel-Gläser) and Immu-Mount mounting medium (Shandon, Thermo Scientific). For experiments with dual mRNA and protein labeling, instead of mounting after the hybridization protocol, the sections were subjected to the IHC as described here. For IHC staining in Fig. 6e,f, the previously described procedure was used[18].

### Antibodies for IHC
The following primary antibodies used for immunocytochemistry were used: anti-Satb2 (1:500, rabbit, homemade[36]), anti-Bcl11b/Ctip2 (1:500, rat, Abcam, 25B6, RRID:AB_2064130), anti-GFP (1:1,000, goat, Rockland Immunochemicals, RRID:AB_2612804), anti-Cre (1:1,000, rabbit, Synaptic Systems, RRID:AB_2619968), anti-Tbr2 (1:300, rabbit, Abcam, RRID:AB_778267), anti-Pax6 (1:500, rabbit, Millipore, RRID:AB_1587367), Draq5 (1:2,000), anti-eIF4EBP1 (1:1,000, rabbit, Abcam, ab32024, RRID:AB_2097990), and anti-phospho-eIF-4EBP1 Thr37/46 (1:1,000, rabbit, Cell Signaling Technology, 2855, RRID:AB_560835). All secondary antibodies were from Jackson ImmunoResearch and were used at a dilution of 1:250.

### Confocal imaging
For Figs. 2, 3 and 6g, imaging of brain coronal cross sections after IUE was performed at the level of primary somatosensory neocortex. For imaging of the overview of immunostaining, a Leica SPL confocal microscope with ×20, ×40 and ×63 objectives was used. For quantitative imaging of the FISH signal, a Leica SP8 microscope with ×40 objective was used. Quantification of mRNA cluster sizes, as well as mRNA and protein localization, was performed using ImageJ software. For Fig. 6e,f, imaging was performed on a Zeiss LSM 880 with ×20 objective, and image stitching was done in Fiji.

### Quantification of mRNA clusters
mRNA puncta were quantified using ImageJ software. The maximum intensity of confocal image Z-stacks was projected on a single two-dimensional plane. After thresholding, the images were binarized using the watershed segmentation to separate cluster clouds. The number of particles of size 0.1 µm$^2$ or larger were then quantified using the Measure Particles tool and normalized to the number of DAPI-labeled nuclei in a given cortical area (for example, VZ and CP). Area of clusters was also quantified and expressed as an absolute surface. Statistical comparison of immediately adjacent neocortical layers from deep (ventricular zone) to superficial was conducted by unpaired two-tailed *t*-test with Welch's correction, or Mann–Whitney *U*-test, after Shapiro–Wilk normality test. See Supplementary Table 3.

## Quantification of colocalization

Mander's colocalization coefficient was quantified for neurons expressing Satb2 and Bcl11b protein and mRNA. Protein colocalization was determined manually, and RNA colocalization was quantified using binarized images after multiplication. Statistical comparison of immediately adjacent neocortical layers from deep (ventricular zone) to superficial was conducted by unpaired two-tailed $t$-test with Welch's correction, or Mann–Whitney $U$-test, after Shapiro–Wilk normality test. See Supplementary Table 3.

## Quantification of neuronal cell markers

The manually quantified number of neurons expressing a given marker was normalized to the entire number of IUE-labeled neurons or to the DAPI-labeled nuclei count. See Supplementary Table 3.

## Statistical analyses

Statistics were performed using SPSS (v17) or GraphPad Prism software. All numerical values and description of statistical tests used, definition of center, dispersion, precision and definition of significance can be found in Supplementary Table 3. Prior to comparison of experimental groups, normality and log-normality tests were performed.

## Reporting summary

Further information on research design is available in the Nature Portfolio Reporting Summary linked to this article.

## Data availability

Data have been deposited in publicly available repositories as indicated: RNA-seq data are publicly available in the NIH GEO under accession number GSE157425, Ribo-seq data are deposited in the NIH GEO under accession number GSE169457, tRNA qPCR array data are deposited in the NIH GEO under accession number GSE169621, and mass spectrometry data are publicly available in the ProteomeXchange under accession number PXD014841. This study further used GENCODE release M12. Source data are provided with this paper.

## Code availability

Code generated during this study is supplied at https://github.com/ohlerlab/cortexomics. Further requests may be directed to and will be fulfilled by the lead contact (M.L.K.).

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

## Acknowledgements

We thank Nadja Klein for advice regarding modeling and statistics, and Heike Heilmann for support with immuno-electron microscopy. M.L.K. thanks James Millonig, Tatiana Popovitchenko, Thorsten Mielke and Martin Vingron for support. M.L.K. was funded by an EMBO Long-Term Postdoctoral Fellowship (190–2016), Alexander von Humboldt Foundation Postdoctoral Fellowship, and an International Guest Fellowship from the Max Planck Institute for Molecular Genetics. D.H. was supported by a grant from the Klaus Tschira Boost Fund. IUE experiments were funded by a Russian Science Foundation Basic Research grant (19-34-51009) and Deutsche Forschungsgemeinschaft grant (DFG TA303/13-1) to V.T.

## Author contributions

M.L.K. designed and initiated the study, with D.H. and M.C.A. making essential contributions. M.L.K., U.O. and V.T. supervised the study, with contributions from C.M.T.S., M.L., M.S. and T.M. Computational experiments were performed by D.H. and G.V., and laboratory experiments were performed by M.C.A., M.L.K., A.R., E.B., A.N.D., M.C.-I, and R.D. Ribo-seq and RNA-seq sample preparation and sequencing were performed by U.Z. and M.L.K., and mass spectrometry sample preparation and measurement were performed by K.I. and M.L.K. Immuno-electron microscopy samples were prepared by A.M.-W. and imaged by B.F. and M.L.K. Data were interpreted by M.L.K., D.H. and M.C.A. Manuscript figures and text were composed by M.L.K., D.H. and M.C.A., with valuable editing and input from all authors.

## Funding

## Competing interests

The authors declare no competing interests.

## Additional information

**Extended data** is available for this paper at https://doi.org/10.1038/s41594-022-00882-9.

**Correspondence and requests for materials** should be addressed to Dermot Harnett, Mateusz C. Ambrozkiewicz or Matthew L. Kraushar.

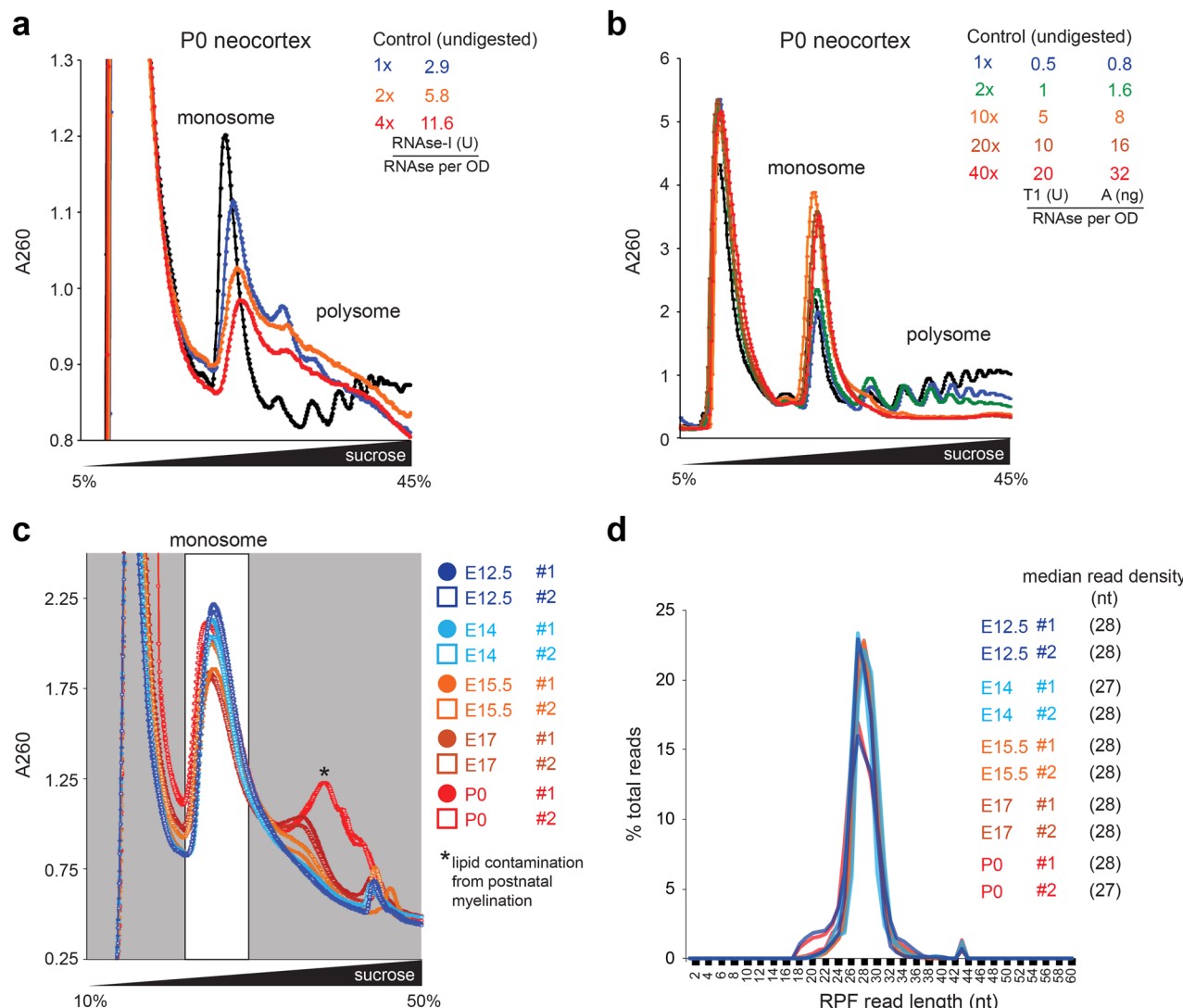

**Extended Data Fig. 1 | Optimized ribosome protected mRNA fragment purification from neocortex.** Nuclease digestion for the generation of ribosome protected mRNA fragments (RPFs) from P0 neocortex, with **a**, RNAse-I vs. **b**, a combination of RNAse-T1 & A. Absorbance at 260 nm (A260). Chains of actively translating ribosomes (polysome) should be digested into single ribosomes (monosome). RNAse-I, typically used in yeast, was inefficient in neocortex lysates, and thus an RNAse-T1 & A protocol was used for this study. **c**, Nuclease digestion and purification of neocortex RPFs in biological duplicates at each developmental stage with the optimized protocol from (b). Each biological replicate included 17–40 brains (34–80 neocortex hemispheres) as detailed in the Methods. **d**, RPF read length distribution. Associated with Fig. 1. See also Supplementary Table 1.

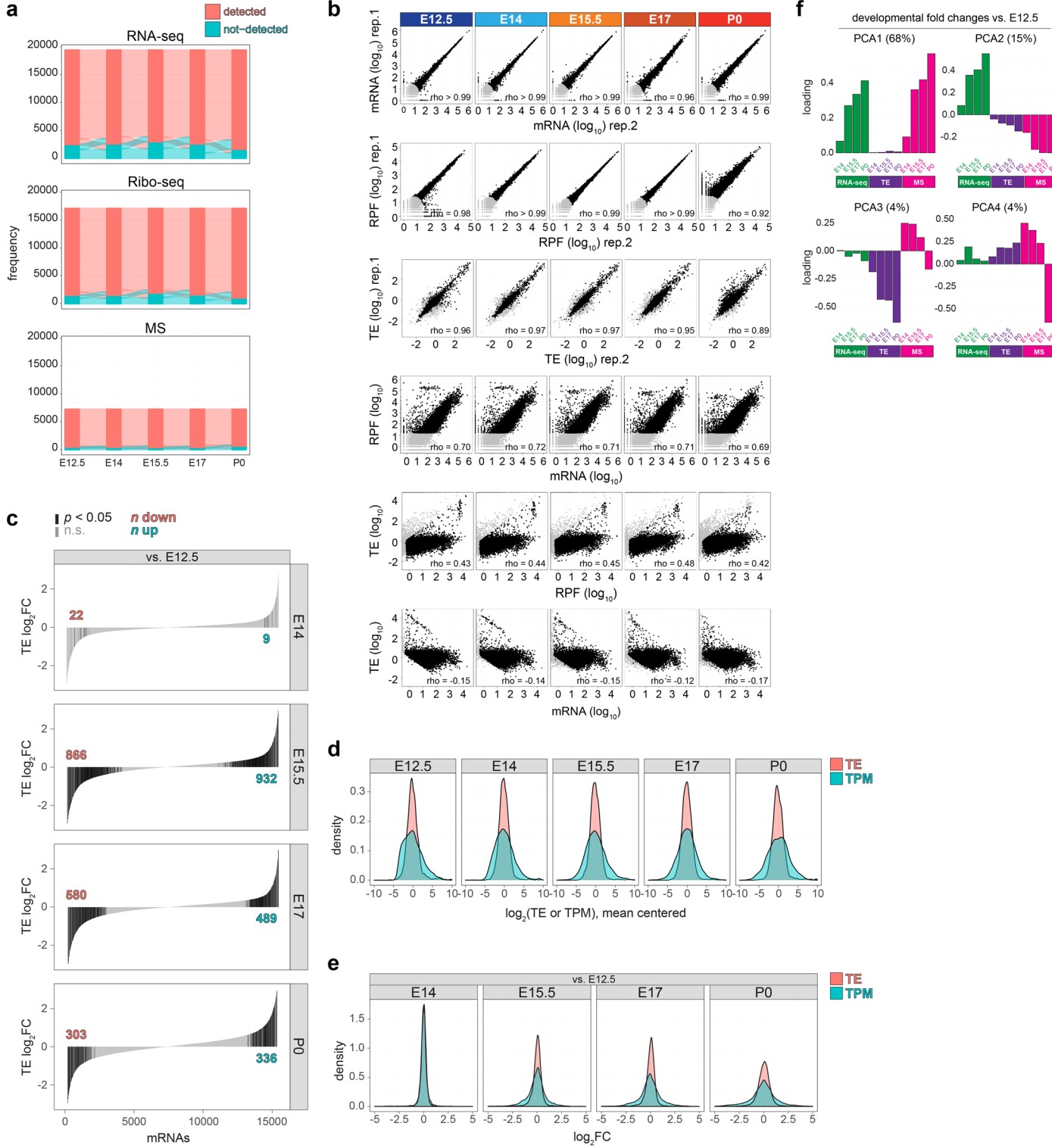

**Extended Data Fig. 2 | Neocortex RNA-seq, Ribo-seq, MS, and translation efficiency data characteristics. a**, River plots demonstrating the number of unique genes detected across all 5 stages measured by RNA-seq, Ribo-seq, or mass spectrometry, compared to the number detected in <5 stages. **b**, Biological replicates of transcripts per million (TPM) measured by RNA-seq (mRNA), Ribo-seq (RPF), and calculated translation efficiency (TE), including correlations between RPF and TE with mRNA to highlight genes with robust translation regulation. **c**, The distribution of TE up and down fold changes (FC) compared to the earliest stage E12.5, with significant genes highlighted in black ($p < 0.05$). Differential expression was called with an empirical Bayes moderated two-sided t-test with adjustment for multiple comparisons. **d**, The distribution of TE and mRNA abundance (TPM) for all genes at each stage, and **e**, fold changes vs. the earliest stage E12.5. **f**, Principal component analysis (PCA) of developmental fold changes in RNA-seq, TE, and MS compared to the earliest stage E12.5. The first four components are shown, with percent variance annotated. Associated with Fig. 1. See also Supplementary Tables 1–2.

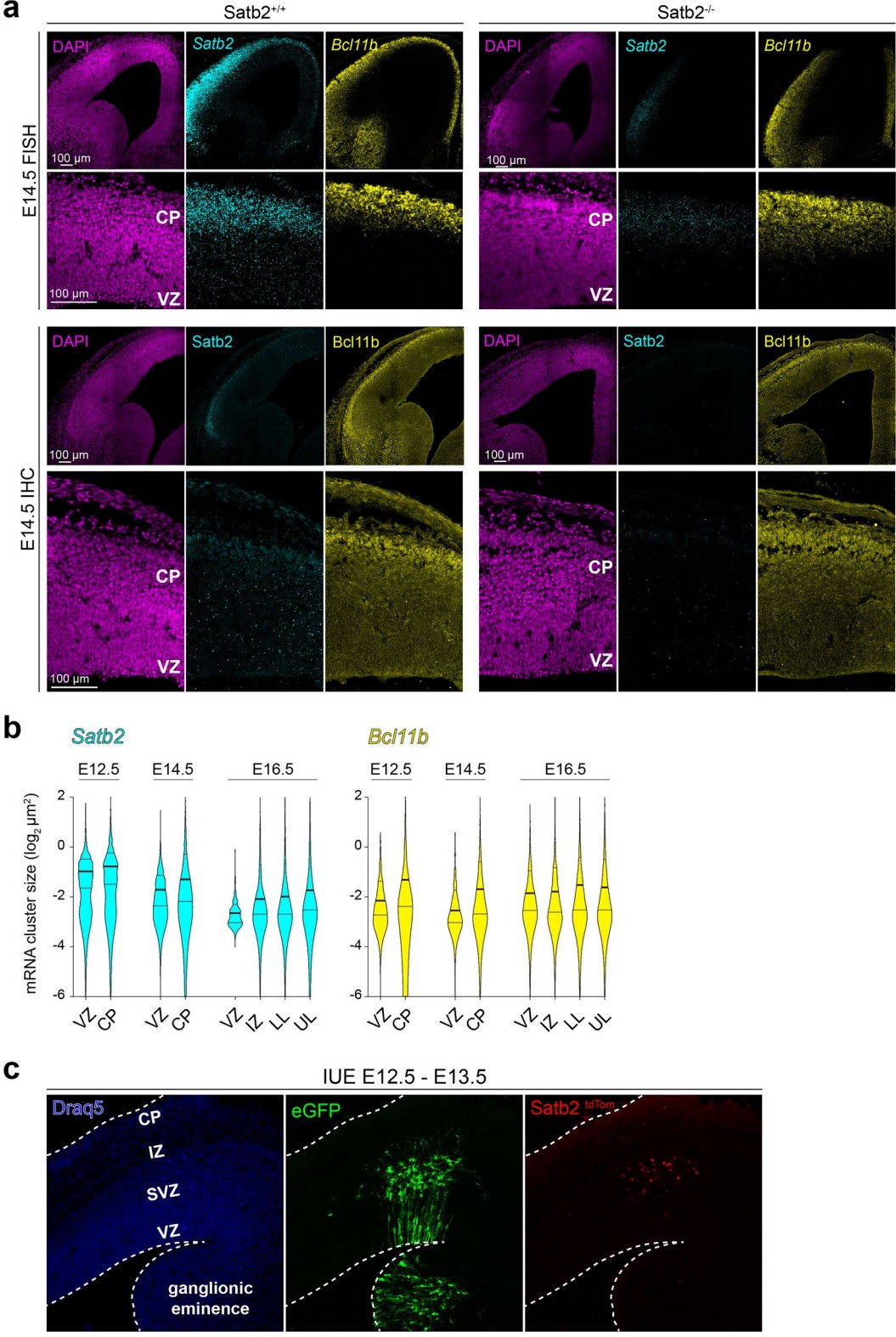

**Extended Data Fig. 3 | *Satb2*[−/−] control for FISH and IHC and neocortex-specific *Satb2* transcription. a**, Fluorescence *in situ* hybridization (FISH) and immunohistochemistry (IHC) probing for *Satb2* and *Bcl11b* mRNA and protein, respectively, in wild-type (*Satb2*[+/+]) and *Satb2* knockout (*Satb2*[−/−]) neocortex coronal sections at E14.5. Ventricular zone (VZ), cortical plate (CP). **b**, Measurement of *Satb2* and *Bcl11b* mRNA cluster sizes in FISH probed neocortex sections at three developmental stages. Intermediate zone (IZ), lower layers (LL), upper layers (UL). Center is median, bounds are quartiles. **c**, *Satb2* transcription activation visualized in *Satb2*[Cre/+] mice by *in utero* co-electroporation of the neocortex and ganglionic eminence with a *loxP-STOP-loxP-tdTomato* (*Satb2*[tdTom]) fluorescence reporter at E12.5, along with eGFP reporter for all transfected cells, and analysis in coronal sections at E13.5. Sub-ventricular zone (SVZ). Associated with Figs. 2 and 3. See also Supplementary Table 3.

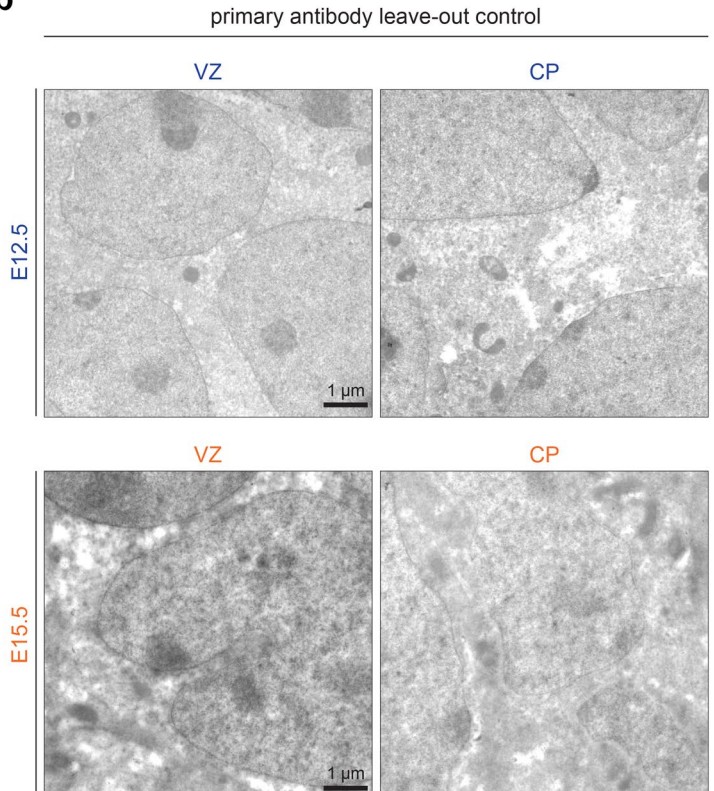

**Extended Data Fig. 4 | See next page for caption.**

**Extended Data Fig. 4 | Immuno-electron microscopy labeling of ribosomes.**
**a**, Raw images of neocortex coronal sections at E12.5 and E15.5 shown in Fig. 4c, immunolabeled with anti-ribosomal protein uS7 followed by 2.5 nm gold secondary (dark black spots), which were automatically detected and quantified in FIJI (magenta spots in Fig. 4c). Electron microscopy was performed in regions corresponding to the stem cell niches of the ventricular zone (VZ) and sub-ventricular zone (SVZ), in addition to regions of differentiating neurons in the cortical plate (CP), which includes both lower layers (LL) and upper layers (UL) at later stages. Quantification of nanogold secondary signal was performed per unit area of the cytoplasm, with nuclei excluded by tracing the nuclear membrane (black lines in Fig. 4c). **b**, Primary antibody leave-out controls were prepared in parallel. Cell images were captured from 2 independent brains, 2 sections/brain, at each developmental stage. Quantification of each image with *n* images quantified is reported in Fig. 4d.

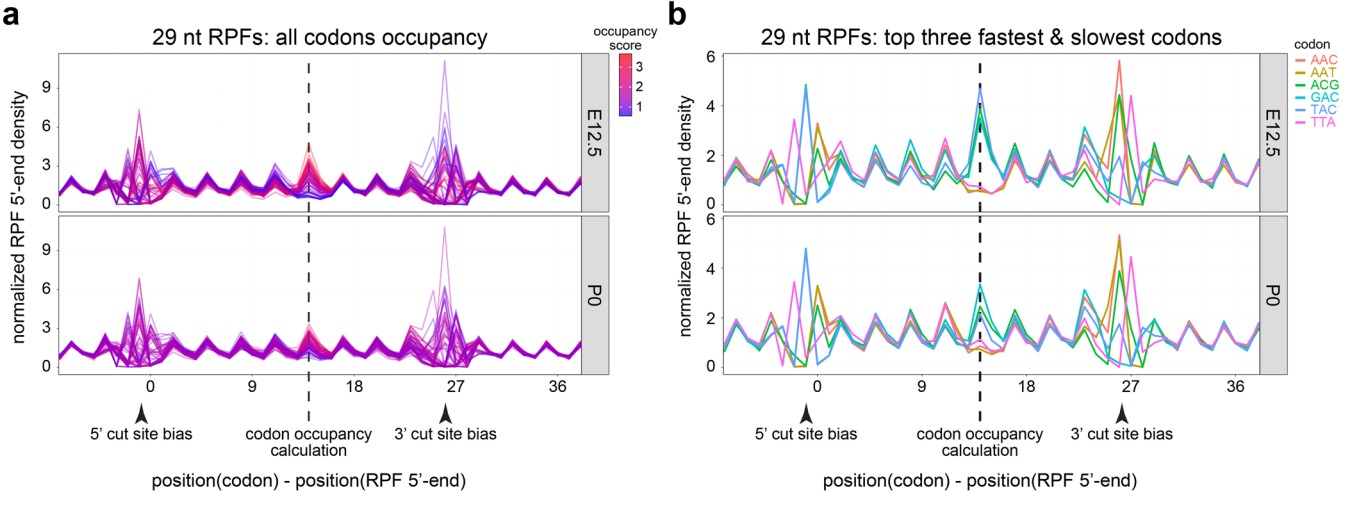

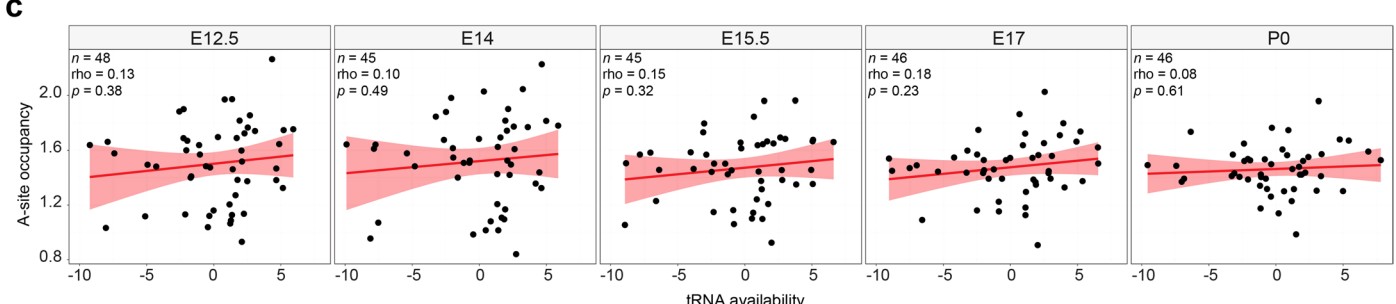

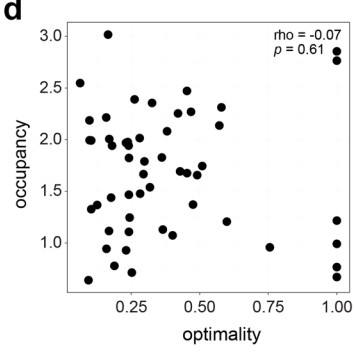

**Extended Data Fig. 5 | Analysis of per-codon ribosome density.** 5′ normalized ribosome-protected mRNA fragment (RPF) density for **a**, all codons and **b**, the top 3 slowest/fastest codons. Plotting the normalized density of Ribo-seq read 5′ ends relative to each codon/read length/sample shows two strongly variable regions corresponding to 5′- and 3′-end cut site biases during nuclease digestion. A third variable region in between corresponds to RPFs with their A/P-sites positioned over the codon. We infer the location of the A-site as the 3 bp region showing the most inter-codon variability (see Methods), and use the normalized occupancy here to measure codon density, and variance between codons.

Independently, this region also identifies the location of intra-codon variability between samples. **c**, Per-codon correlation between tRNA availability calculated from tRNA qPCR array (see Methods), and the ribosome occupancy of that codon when positioned in the A-site of the ribosome footprint. Center is maximum likelihood slope, ribbon is 95% CI. **d**, Correlation between ribosome occupancy per codon and the optimality of the codon as defined in ref. [46], with the mean across all stages shown. Association between paired samples in (c, d) was tested with Pearson's product moment correlation coefficient. Associated with Fig. 5. See also Supplementary Table 4.

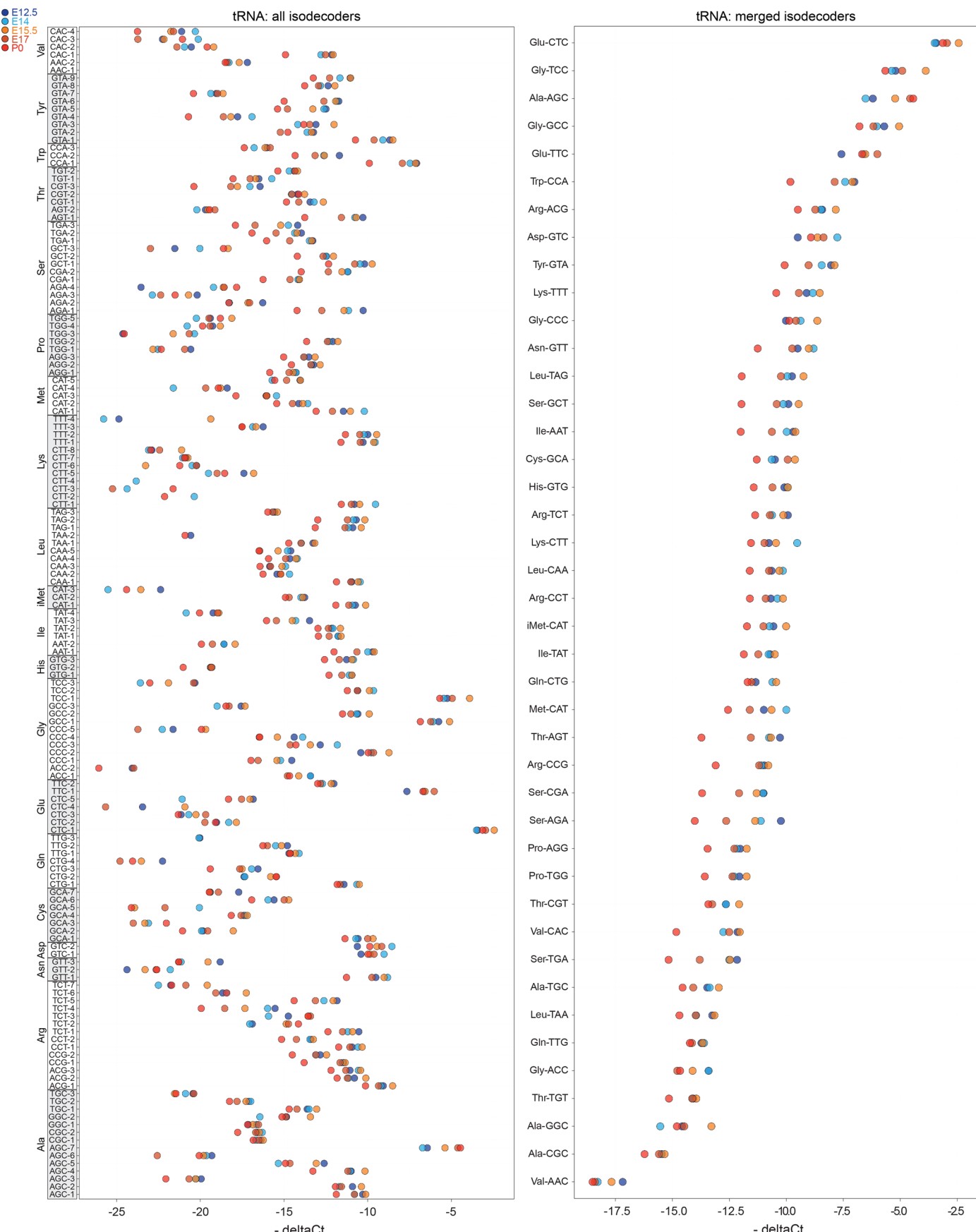

**Extended Data Fig. 6 | Neocortex tRNA qPCR array.** Total tRNA levels at each stage measured by qPCR array in biological duplicate, with Ct values for each tRNA isodecoder (left) or averaged across isodecoders (right) compared to the mean of 5S and 18S rRNA levels in each sample (delta Ct). Associated with Fig. 5. See also Supplementary Table 4.

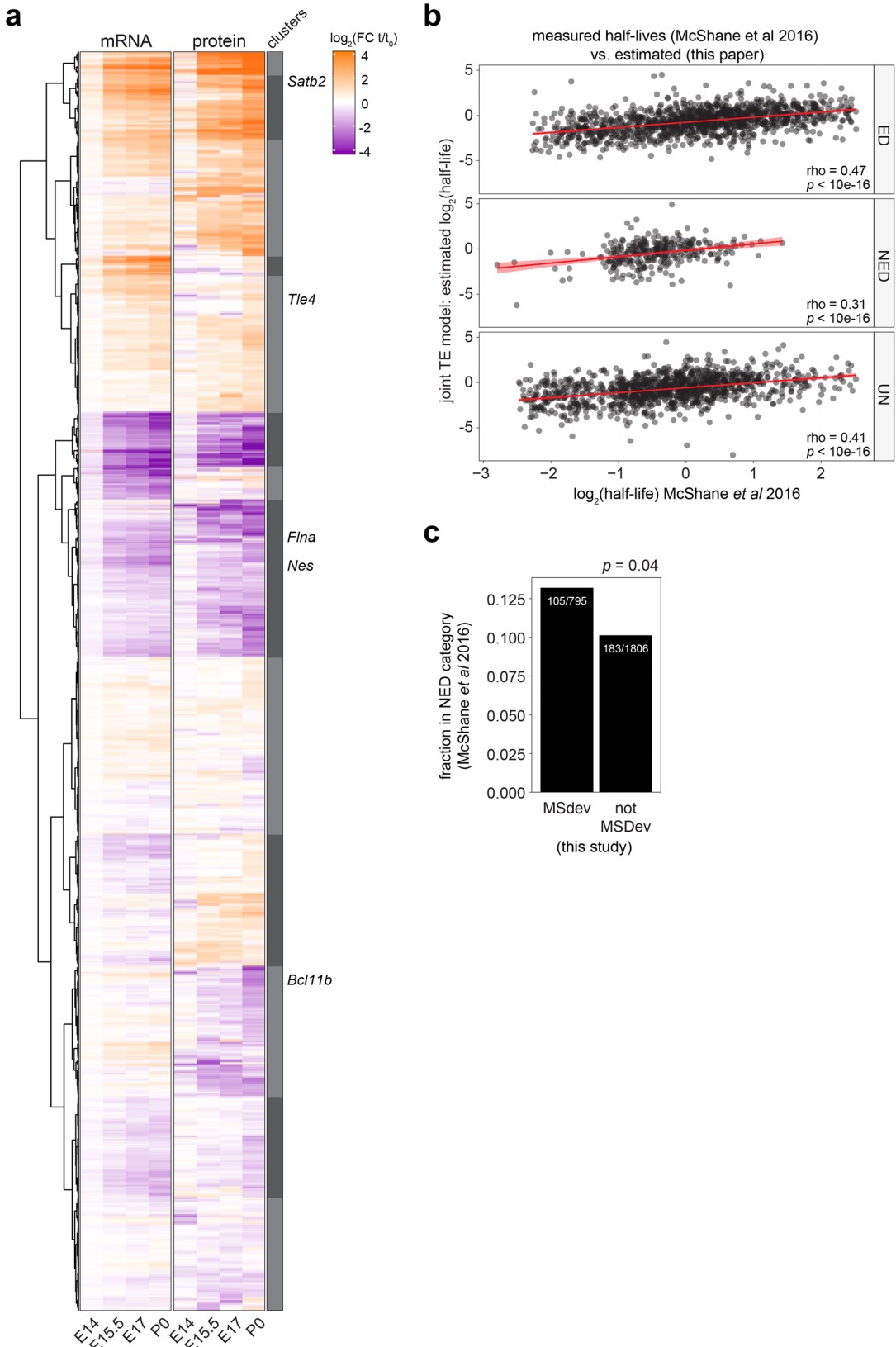

**Extended Data Fig. 7 | Modeling of mRNA translation. a**, Hierarchical clustering based on mRNA (RNA-seq) and protein (MS) expression trajectories per gene. Fold change expression increasing or decreasing from E12.5 ($t_0$) to subsequent developmental stages (t) shown in heat map. Neural stem cell and neuronal marker genes are indicated (right). **b**, Protein half-lives measured by SILAC MS and categorized as exponential decay (ED), non-exponential decay (NED), or neither (UN) in[57] correlated with the model estimates from our data as per[56]. Center is maximum likelihood slope, ribbon is 95% CI. **c**, The fraction of genes modeled as MS deviating or non-deviating in this study that are categorized as NED proteins in[57]. Fisher's exact test for significance. Associated with Fig. 7. See also Supplementary Table 5.

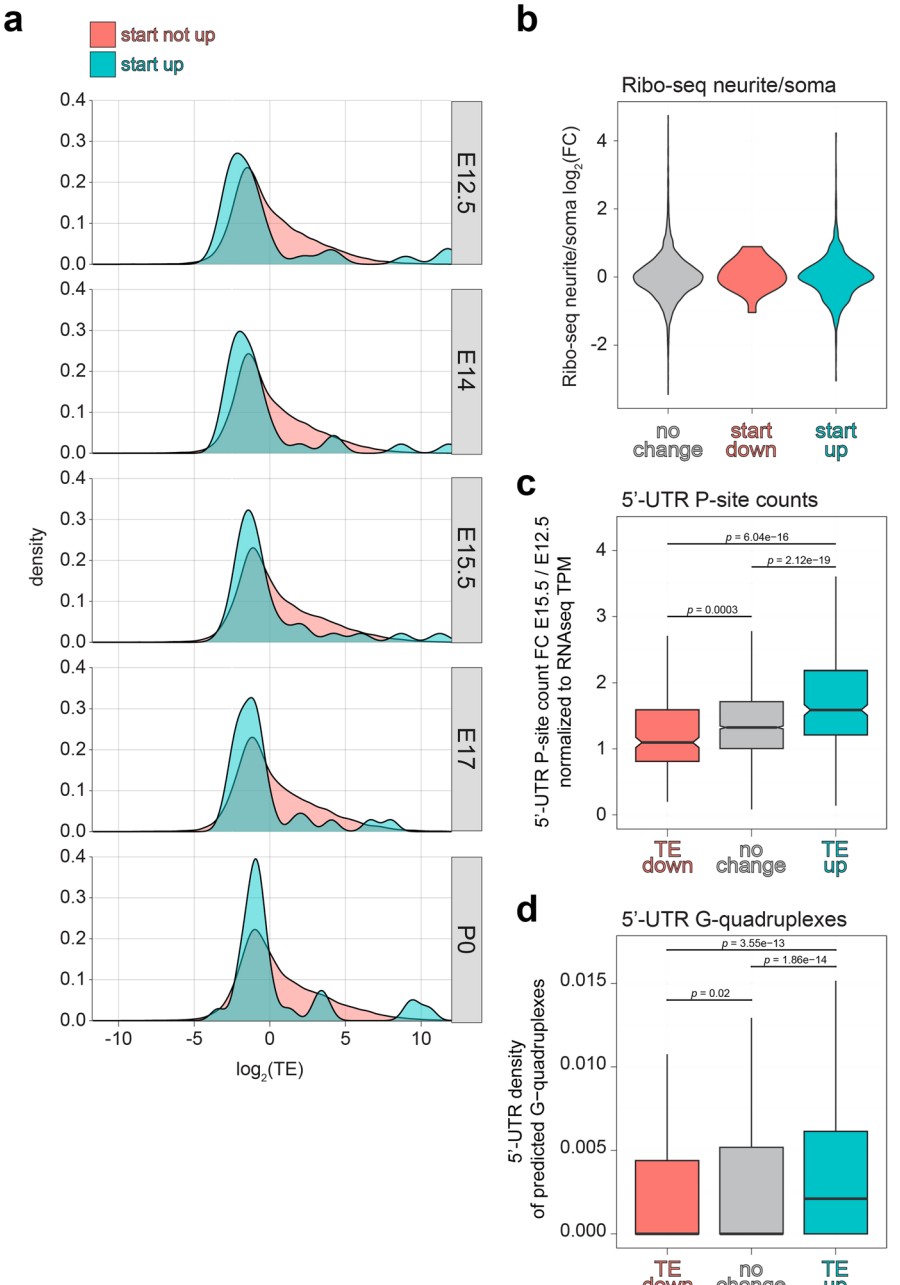

**Extended Data Fig. 8 | Start codon effect and 5′-UTR analysis. a**, Translation efficiency (TE) distribution for genes with increasing start codon occupancy across developmental stages, compared to those without start occupancy changes. **b**, The association of mRNAs demonstrating start codon occupancy changes with translation in neurites vs. the soma of cultured neurons[98]. **c**, Change in ribosome density from E12.5-E15.5 upstream of the main open reading frame for TE down vs. up mRNAs. **d**, Density of predicted G-quadruplex-forming sequences[94] in the 5′-UTR of TE down vs. up mRNAs. Significant differences between groups in (c, d) assessed by unpaired two-tailed Wilcoxon test, $n = 1129$ TE up, 1131 TE down, and 9968 no TE change genes calculated from Ribo-seq and RNA-seq analyses of 2 biologically independent neocortex lysates. Boxplot centers are median, with quartile bounds, and whiskers are observations ≥ or ≤ hinges + or - 1.5 * IQR, respectively. See Main and Methods text for details. Associated with Fig. 5.

| | |
|---|---|

# Reporting Summary

## Statistics

For all statistical analyses, confirm that the following items are present in the figure legend, table legend, main text, or Methods section.

| n/a | Confirmed | |
|---|---|---|
| ☐ | ☒ | The exact sample size (*n*) for each experimental group/condition, given as a discrete number and unit of measurement |
| ☐ | ☒ | A statement on whether measurements were taken from distinct samples or whether the same sample was measured repeatedly |
| ☐ | ☒ | The statistical test(s) used AND whether they are one- or two-sided<br>*Only common tests should be described solely by name; describe more complex techniques in the Methods section.* |
| ☐ | ☒ | A description of all covariates tested |
| ☐ | ☒ | A description of any assumptions or corrections, such as tests of normality and adjustment for multiple comparisons |
| ☐ | ☒ | A full description of the statistical parameters including central tendency (e.g. means) or other basic estimates (e.g. regression coefficient) AND variation (e.g. standard deviation) or associated estimates of uncertainty (e.g. confidence intervals) |
| ☐ | ☒ | For null hypothesis testing, the test statistic (e.g. *F*, *t*, *r*) with confidence intervals, effect sizes, degrees of freedom and *P* value noted<br>*Give P values as exact values whenever suitable.* |
| ☐ | ☒ | For Bayesian analysis, information on the choice of priors and Markov chain Monte Carlo settings |
| ☐ | ☒ | For hierarchical and complex designs, identification of the appropriate level for tests and full reporting of outcomes |
| ☐ | ☒ | Estimates of effect sizes (e.g. Cohen's *d*, Pearson's *r*), indicating how they were calculated |

*Our web collection on statistics for biologists contains articles on many of the points above.*

## Software and code

Policy information about availability of computer code

| Data collection | No commercial, open source, or custom code was utilized for data collection. |
|---|---|
| Data analysis | For data analysis, code generated during this study is supplied at: https://github.com/ohlerlab/cortexomics. Further requests may be directed to and will be fulfilled by the Lead Contact, matthew.kraushar@molgen.mpg.de (M.L.K.). Code or software packages derived from previously published work are described with referenced sources in the Methods, which includes the following: Bcl2fastq 2.20.0, STAR 2.6.1a, Limma, Xtail 1.1.5, ProDA, RUST, DeepShapePrime, Salmon 0.14.1, Snakemake, Tximport, Voom, RiboDiPA, TopGO, MaxQuant v1.6.0.1, NanoString, Hclust, AME/Meme suite 5.1.1, Pqsfinder 2.10.1, Cutadapt 1.18, Graph Pad Prism 7, FIJI, IBAQ, LFQ, Perseus, R v4.0.0, SPSS v.17, Ribo-seQC |

For manuscripts utilizing custom algorithms or software that are central to the research but not yet described in published literature, software must be made available to editors and reviewers. We strongly encourage code deposition in a community repository (e.g. GitHub). See the Nature Portfolio guidelines for submitting code & software for further information.

## Data

Policy information about availability of data

All manuscripts must include a data availability statement. This statement should provide the following information, where applicable:
- Accession codes, unique identifiers, or web links for publicly available datasets
- A description of any restrictions on data availability
- For clinical datasets or third party data, please ensure that the statement adheres to our policy

Data have been deposited in publicly available repositories as indicated:
RNA-seq data are publicly available in the NIH GEO: GSE157425.
Ribo-seq data are deposited in the NIH GEO: GSE169457.

# Field-specific reporting

Please select the one below that is the best fit for your research. If you are not sure, read the appropriate sections before making your selection.

☒ Life sciences　　☐ Behavioural & social sciences　　☐ Ecological, evolutionary & environmental sciences

For a reference copy of the document with all sections, see nature.com/documents/nr-reporting-summary-flat.pdf

# Life sciences study design

All studies must disclose on these points even when the disclosure is negative.

| | |
|---|---|
| Sample size | No sample-size calculation was performed. All replicates reported constitute biological replicates. The sample-size for biological replicates was determined based on standards in the fields of RNA sequencing & Ribosome Profiling (Ingolia et al. Science. (324) 2009), tRNA analysis (Zheng et al. Nat Methods. (12) 2015), and mass spectrometry (Tabb et al. J Proteome Res. (9) 2010), which includes the extent of practicable feasibility and financial limitations of replicate measurements. As described in detail in the Methods, this includes biological replicates n = 2 for RNA-seq, Ribo-seq, tRNA qPCR array; n = 3 for mass spectrometry. All biological replicates and sample-sizes for quantification of fluorescent in situ hybridization, immunohistochemistry, and in utero electroporation constitute n ≥ 3, and were determined in part by the availability and financial cost of replicate animals and reagents - all such n's are detailed in the Methods and Supplementary Table 3. |
| Data exclusions | No data were excluded from analyses. |
| Replication | Reproducibility for all experiments were assessed statistically. Consistency of our findings, including significant differences between conditions, were tested with the reported statistical tests in the Methods, Figure Legends, and Supplementary Tables. All numerical values and description of statistical tests used, definition of center, dispersion, precision, and definition of significance can be found in the Supplementary Tables and in the Methods. Prior to comparison of experimental groups, normality and log-normality test were performed. Reproducibility of non-quantitative data, such as fluorescence or electron microscopy images, were confirmed across multiple replicate images derived from independent animals and electroporations. |
| Randomization | Mice were utilized in the embryonic (E12.5-E17) and early post-natal (P0) period, inclusive of both male and female sexes in each litter without distinction. Sequencing (n = 2), qPCR-array (n = 2), and mass spectrometry (n = 3) experiments pooled multiple animals' cortex tissue across multiple independent litters for each biological replicate. The assignment of wild type animals into omics, FISH, and IHC experimental groups was not randomized, but simply followed their developmental stage. For in utero electroporation experiments, animals within the same litter either received control or overexpression plasmids; thus while not randomized, the proper intervention-control comparisons were performed appropriate for statistical comparison. |
| Blinding | Investigators were blinded for both imaging and quantification of immunohistochemistry, in situ hybridization, and in utero electroporation experiments, as the investigators themselves were responsible for deriving quantitative data from microscopy image data. For computational bioinformatic analysis of sequencing, qPCR array, and mass spectrometry data, investigators were not blinded to sample condition. We justify the bioinformatics being un-blinded because the computational algorithms/scripts were written in advance of the analysis and quantification, and were run exactly the same between all experimental groups. |

# Reporting for specific materials, systems and methods

We require information from authors about some types of materials, experimental systems and methods used in many studies. Here, indicate whether each material, system or method listed is relevant to your study. If you are not sure if a list item applies to your research, read the appropriate section before selecting a response.

## Materials & experimental systems

| n/a | Involved in the study |
|---|---|
| ☐ | ☒ Antibodies |
| ☒ | ☐ Eukaryotic cell lines |
| ☒ | ☐ Palaeontology and archaeology |
| ☐ | ☒ Animals and other organisms |
| ☒ | ☐ Human research participants |
| ☒ | ☐ Clinical data |
| ☒ | ☐ Dual use research of concern |

## Methods

| n/a | Involved in the study |
|---|---|
| ☒ | ☐ ChIP-seq |
| ☒ | ☐ Flow cytometry |
| ☒ | ☐ MRI-based neuroimaging |

## Antibodies

| | |
|---|---|
| Antibodies used | As described in the Methods, primary antibodies used for immunocytochemistry were used at dilutions indicated: anti-Satb2 (1:500, rabbit, home-made; Ambrozkiewicz et al. J Neurosci Methods. (291) 2017), anti-Bcl11b/Ctip2 (1:500, rat, Abcam, 25B6, |

RRID:AB_2064130), anti-GFP (1:1000, goat, Rockland, RRID:AB_2612804), anti-Cre (1:1000, rabbit, SySy, RRID:AB_2619968), anti-Tbr2 (1:300, rabbit, Abcam, RRID:AB_778267), anti-Pax6 (1:500, rabbit, Millipore, RRID:AB_1587367), Draq5 (1:2000), anti-eIF4EBP1 (1:1000, rabbit, Abcam, ab32024, RRID:AB_2097990), anti-phospho-eIF4EBP1 Thr37/46 (1:1000, rabbit, Cell Signaling, 2855, RRID:AB_560835), anti-Rps5 a.k.a. uS7 (1:1000, mouse, Santa Cruz, sc-390935, RRID:AB_2713966). All secondary Cy2, Cy3, and Cy5 antibodies were from Jackson Immunoresearch and were used at 1:250.

Primary antibodies used for Western blot were used at the dilutions indicated: anti-eIF4EBP1 (1:1000, rabbit, Abcam, ab32024, RRID:AB_2097990), anti-phospho-eIF4EBP1 Thr37/46 (1:1000, rabbit, Cell Signaling, 2855, RRID:AB_560835), anti-Gapdh (1:1000, mouse, Millipore, MAB374, RRID:AB_2107445). All Horse Radish Peroxidase secondary antibodies were used at 1:2500: HRP-anti-mouse-Heavy Chain (goat, Abcam, ab97245; RRID:AB_10680049), HRP-anti-rabbit-Heavy Chain (goat, Cell Signaling, 7074S, RRID:AB_2099233).

| Validation | The antibody anti-Satb2 (rabbit, home-made; Ambrozkiewicz et al. J Neurosci Methods. (291) 2017) was validated by immunohistochemistry in Satb2 -/- (knockout) brains as described in the Extended Data. The antibody anti-Rps5 a.k.a. uS7 (mouse, Santa Cruz, sc-390935, RRID:AB_2713966) was further validated by western blot probing of total neocortex lysates across all developmental stages (E12.5-P0) in a prior study from the authors (Kraushar et al. Mol Cell. (81) 2021), demonstrating a clear single band at the appropriate molecular weight. Otherwise, we did not independently validate the other antibodies used; beyond what is annotated manufacturers' standard methods, which include western blot testing of genomic knock-out lysates, and/or in comparison to recombinantly expressed protein corresponding to the immunogen. We defer to the manufacturer-specific details, which are searchable at the catalogue IDs and RRIDs provided for all antibodies used in our study. |
|---|---|

# Animals and other organisms

Policy information about studies involving animals; ARRIVE guidelines recommended for reporting animal research

| Laboratory animals | Mice were housed in 12 hour light/12 hour dark cycle, at a consistent 18-23 °C, 40-60% humidity, with pellet food and water available ad libitum. Mice were utilized in the embryonic (E12.5-E17) and early post-natal (P0) period with the stage and replicate numbers as reported for each experiment. Each sample was inclusive of both male and female sexes in each litter without distinction. Timed pregnant wild-type (WT) CD-1 mice utilized for Ribo-seq, RNA-seq, tRNA qPCR array, mass spectrometry, and immuno-electron microscopy were obtained from the Charles River Company (Protocol: T0267/15). Experiments with fluorescent in situ hybridization and immunohistochemistry were performed in NMRI WT mice. For experiments with the tdTomato reporter, Satb2Cre/+ males (Britanova et al. Neuron. (57) 2008) were mated to NMRI wild type females (Protocols: G0079/11, G54/19, and G206/16). Satb2Cre/+ mouse genotyping was performed as described (Britanova et al. Neuron. (57) 2008). |
|---|---|
| Wild animals | The study did not involve wild animals. |
| Field-collected samples | The study did not involve animals collected from the field. |
| Ethics oversight | Mouse (Mus musculus) lines were maintained in the animal facilities of the Charité University Hospital and Lobachevsky State University. All experiments were performed in compliance with the guidelines for the welfare of experimental animals approved by the State Office for Health and Social Affairs, Council in Berlin, Landesamt für Gesundheit und Soziales (LaGeSo), permissions T0267/15, G0079/11, G206/16, and G54/19, and by the Ethical Committee of the Lobachevsky State University of Nizhny Novgorod. |

Note that full information on the approval of the study protocol must also be provided in the manuscript.

