## [Peer Review File · Nature Structural & Molecular Biology]

Peer Review Information

Journal: Natural Structural & Molecular Biology

Manuscript Title: A critical period of translational control during brain development at codon resolution

Corresponding author name(s): Matthew L. Kraushar, Mateusz C. Ambrozkiwicz, Dermot Harnett

Reviewer Comments & Decisions:

Decision Letter, initial version:
--

Message: 24th Feb 2022

Dear Dr. Kraushar,

Thank you again for submitting your manuscript "A critical period of translational control during brain development at codon resolution". I apologize for the delay in responding, which resulted from the difficulty in obtaining suitable referee reports. Nevertheless, we now have comments (below) from the 2 reviewers who evaluated your paper. In light of those reports, we remain interested in your study and would like to see your response to the comments of the referees, in the form of a revised manuscript.

Please be sure to address/respond to all concerns of the referees in full in a point-by-point response and highlight all changes in the revised manuscript text file. If you have comments that are intended for editors only, please include those in a separate cover letter.

We expect to see your revised manuscript within 6 weeks. If you cannot send it within this time, please contact us to discuss an extension; we would still consider your revision, provided that no similar work has been accepted for publication at NSMB or published elsewhere.

As you already know, we put great emphasis on ensuring that the methods and statistics reported in our papers are correct and accurate. As such, if there are any changes that should be reported, please submit an updated version of the Reporting Summary along

with your revision.

Reporting Summary:

When submitting the revised version of your manuscript, please pay close attention to our [href="https://www.nature.com/nature-research/editorial-policies/image-integrity">Digital Image Integrity Guidelines. and to the following points below:](https://www.nature.com/nature-research/editorial-policies/image-integrity)

Please note that all key data shown in the main figures as cropped gels or blots should be presented in uncropped form, with molecular weight markers. These data can be aggregated into a single supplementary figure item. While these data can be displayed in a relatively informal style, they must refer back to the relevant figures. These data should be submitted with the final revision, as source data, prior to acceptance, but you may want to start putting it together at this point.

Data availability: this journal strongly supports public availability of data. All data used in accepted papers should be available via a public data repository, or alternatively, as Supplementary Information. If data can only be shared on request, please explain why in your Data Availability Statement, and also in the correspondence with your editor. Please note that for some data types, deposition in a public repository is mandatory - more information on our data deposition policies and available repositories can be found below:

<https://www.nature.com/nature-research/editorial-policies/reporting-standards#availability-of-data>

[Redacted]

Sincerely,

Sara Osman, Ph.D.
Associate Editor
Nature Structural & Molecular Biology

Referee expertise:

Referee #1: Cortical development

Referee #2: Functional genomics, neurogenesis

Reviewers' Comments:

Reviewer #1:

Remarks to the Author:

In this study, Harnett et al. show the dynamics between transcription and translation during neurogenesis of the mouse neocortex using RNA-seq, Ribo-seq and MS. The authors show that the timed translational regulation of *Satb2*, despite having mRNA expression during development, restricts the protein expression in neuronal lineage. Mechanistically, eIF4EBP1 expression is differentially regulated to decrease ribosome protein levels, coinciding with lower ribosome numbers during late neurogenesis. Therefore, this study demonstrates how broad transcriptional programs can be regulated by translation during mouse neocortical development.

The manuscript is well written, and the work presented here is of high quality. Additionally, the transcriptome of mouse neocortical development generated in this study will be a valuable resource for the cortical development community for years to come (I have, however, not been able to access the full website). Although most conclusions are well supported by the data, there are a few remaining points that need to be addressed, as outlined below.

- 1) In figure 2B, the authors show the RNA-seq, Ribo-seq and MS profile of *Satb2*, concluding that there is a 2-fold increase in translation due to the higher protein abundance. However, in figure 2D, the protein abundance is much higher compared to earlier stages and not reflective of an overall 2-fold increase. Could this mean that the MS profiling underestimates the actual dynamics of the protein? If so, the authors are advised to mention this caveat in the manuscript. Additionally, if the authors can provide examples of other genes/proteins that have similar dynamics as *Satb2* in Fig. 2B, it would help support the method of cross-examination of RNA-seq/Ribo-seq/MS datasets used.
- 2) In figure 2E, the authors have quantified the in-situ and immunostaining signal presented in figure 2D. However, some data points only have two replicates, and no statistical analysis is performed. If possible, the dataset should be expanded and quantified.
- 3) In lines 170-175, the authors do not mention a figure annotation which makes it hard for the reader to follow. Additionally, '*Satb2*' should be in italics in line 172.
- 4) In figure 3B, the authors show that transcriptional priming of *Satb2* using a flox-stop-flox-tdTomato plasmid electroporated in the *Satb2*Cre cortices. It would be helpful if the authors made it clearer in the text or the schematic in fig. 3A that in this model, *Satb2* and Cre mRNA are expressed under the control of *Satb2* promoter, whereas only Cre is translated but not *Satb2*, due to the differential protein translation outlined in fig. 2E. That is why Cre can be observed as soon as transcription takes place whereas *Satb2* is absent.
- 5) In figure 3E, the authors show protein expression for Cre, *Satb2* and *Bcl11b*. I appreciate the added quantification of the immunostaining. However, no statistics were performed, and one of the bars only has two datapoints. Again if possible, the dataset should be expanded and quantified.
- 6) In figure 6E, the immunostaining data presented is not very convincing and looks like background signal. This is also highlighted with discrepancy between low eIF4EBP1 P0 western blot signal but high eIF4EBP1 P0 immunostaining if the signal of eIF4EBP1 protein is to be believed. This should be clarified.
- 7) In figure 6G, the authors show a decrease in GFP+ cells in the cortical plate in

eIF4EBP1 electroporations but quantifications do not reflect the image presented in figure 6F. A more representative image would help demonstrate the data more fairly.

8) In figure 7C and lines 376-380, the clusters are mentioned out of order to the written texts. The authors are advised to rearrange the text according to the figure.

Reviewer #2:

Remarks to the Author:

Harnett and Ambrozkiwicz et al studied how gene expression changes during cortical development in the mouse, with an emphasis on post-transcriptional gene regulation. The authors perform a series of unbiased measurements to globally assess gene expression during mouse cortical development. By comparing datasets, the authors identify genes that are selectively translationally controlled. The authors find that *Satb2* is preferentially translated and ribosomal proteins are translationally repressed across development. The work is interesting and describing how gene expression is controlled during brain development is important. The multidisciplinary approach is fantastic and adds depth to the paper and demonstrates robustness of conclusions. Some lines of investigation stop short of exploring all the molecular details of the regulation identified, but that can be developed in future work as this is already a substantial contribution.

Major comments:

The introduction contains a large number of findings from this work in the last two paragraphs. In my opinion, these specific findings should be moved to the results or discussion sections so that the introduction can focus on setting the stage for the work.

The authors discuss the role of 4EBP1 at length, and also discuss 5' TOP dependent translational control, but don't discuss the mTOR signaling pathways that phosphorylate 4EBP1 to regulate 5' TOP translation. The authors also don't measure phosphorylation of 4EBP1 or other mTOR targets. In my opinion, this is an oversight in the work – especially since there is prior literature on the role of mTOR signaling in neuronal differentiation in human and mouse (e.g. PMID 29789464, 32115408, 29141229, 32876565, and more). At minimum, discussing this possibility would enrich the paper, and if the authors wanted to they could perform additional experiments to address these aspects. However, to be clear, this work is already very comprehensive and showing that 4EBP1 phosphorylation is important in this context is definitely beyond the scope of this manuscript.

A collection of mRNAs that are translationally controlled are identified in this work. The authors seek to then define what sequences in these mRNAs might be determinants of translation. The authors do this by identifying enriched sequence motifs in different sets of mRNAs, which is a strong analysis. However, this approach is unlikely to find some important regulatory elements, such as secondary structures and upstream start codons – both of which can be potent translational regulators. The authors have generated a large amount of data and the paper would benefit from slightly deeper exploration of these features.

Minor comments:

Ribosome profiling measures ribosome occupancy which may or may not be a direct measure of active translation since ribosomes can also stall, especially in neurons (e.g.

PMID 24043809). The authors suggest that it is a direct measure of active translation, and clarification might improve the presentation of the technique.

Author Rebuttal to Initial comments

NSMB-RS45687 Revision

Response to Reviewers

Harnett & Ambrozkiwicz *et al.*

“A critical period of translational control during brain development at codon resolution”

Responses in italics

Reviewer #1:

Remarks to the Author:

In this study, Harnett et al. show the dynamics between transcription and translation during neurogenesis of the mouse neocortex using RNA-seq, Ribo-seq and MS. The authors show that the timed translational regulation of *Satb2*, despite having mRNA expression during development, restricts the protein expression in neuronal lineage. Mechanistically, eIF4EBP1 expression is differentially regulated to decrease ribosome protein levels, coinciding with lower ribosome numbers during late neurogenesis. Therefore, this study demonstrates how broad transcriptional programs can be regulated by translation during mouse neocorticalogenesis.

The manuscript is well written, and the work presented here is of high quality. Additionally, the transcriptome of mouse neocortical development generated in this study will be a valuable resource for the cortical development community for years to come (I have, however, not been able to access the full website). Although most conclusions are well supported by the data, there are a few remaining points that need to be addressed, as outlined below.

1) In figure 2B, the authors show the RNA-seq, Ribo-seq and MS profile of *Satb2*, concluding that there is a 2-fold increase in translation due to the higher protein abundance. However, in figure 2D, the protein abundance is much higher compared to earlier stages and not reflective of an overall 2-fold increase. Could this mean that the MS profiling underestimates the actual dynamics of the protein? If so, the authors are advised to mention this caveat in the manuscript. Additionally, if the authors can provide examples of other genes/proteins that have similar dynamics as *Satb2* in Fig. 2B, it would help support the method of cross-examination of RNA-seq/Ribo-seq/MS datasets used.

*We are grateful for this comment, which led us to more clearly detail in the text the comparison of our bioinformatics data with our in situ data in **Figure 2**, and emphasize the difference between translation efficiency fold change and MS fold change.*

*In the improved description, we focus on the acute fold changes between adjacent stages E14 to E15.5 in lines 143-145 (to emphasize the change after E14.5 shown in **Fig. 2d**). We calculate a 1.4-fold increase in translation efficiency of Satb2 mRNA between E14 - E15.5 (plotted as $\log_2(\text{fold change})=0.45$ in **Fig. 2b**); where translation efficiency = Ribo-seq signal / RNA-seq signal. However, for Satb2 protein we calculate an 8.2-fold increase measured by MS between E14 - E15.5 (plotted as $\log_2(\text{fold change})=3$ in **Fig. 2b**). New lines 143-145:*

“In contrast, from E14 to E15.5 Satb2 Ribo-seq signal increases 7.4-fold and MS signal 8.2-fold, in excess of the 5.4-fold change in RNA-seq, yielding a 1.4-fold increase in translation efficiency between these developmental stages.”

*Indeed, an 8.2-fold increase in Satb2 protein measured by MS is more reflective of the substantial increase Reviewer #1 observed by immunofluorescence in **Fig. 2d**. The distinction between fold change in calculated translation efficiency vs. fold change in MS is an important one as indicated by Reviewer #1, and thus we further emphasize this point when reporting the immunohistochemistry in lines 168-170:*

*“Only by E16.5 is Satb2 protein expression robust, concordant with an 8.2-fold increase in MS signal and 1.4-fold upregulation of Satb2 translation efficiency described above (**Fig. 2d**, bottom panels).”*

Of note: the above comment from Reviewer #1 is also relevant to minor comment #1 from Reviewer #2, where we now discuss the distinction between translation efficiency and “ribosome density” – a more appropriate term.

To highlight other genes that show similar dynamics to Satb2, we perform clustering of different expression trajectories, which we describe in lines 381-383:

“Furthermore, several essential neural stem cell and differentiation markers segregate into distinct clusters, such as Nes in cluster J and Satb2 in cluster M”

*We direct the reader in this paragraph to **Supplementary Table 5**, which lists all genes falling into the same cluster “M” as Satb2. Likewise, we now direct the reader in lines 407-410 to all genes showing similar “production” dynamics to Satb2 in the kinetic modeling:*

“Genes in the five modeled categories showed distinct gene ontology term enrichment, such as a linear relationship between the translation and abundance of ribosome components, or the non-linear (production) relationship for chromatin associated proteins, including Satb2 (Fig. 7d, Supplementary Table 5).”

Finally, to point to the data for all genes that are overall translationally upregulated (like Satb2) and downregulated (like ribosomal proteins), we now direct the reader in lines 124 and 220, respectively, to “(Supplementary Table 2)”.

2) In figure 2E, the authors have quantified the in-situ and immunostaining signal presented in figure 2D. However, some data points only have two replicates, and no statistical analysis is performed. If possible, the dataset should be expanded and quantified.

*We have expanded quantification in **Figure 2e**, with now n^3 3 per FISH and IHC experiment. We further perform statistical comparison of adjacent cortical layers starting from deep (ventricular zone) to superficial. We include this description in the new **Figure 2e** legend and **Methods** sections, in addition to reporting the new values in **Supplementary Table 3**.*

3) In lines 170-175, the authors do not mention a figure annotation which makes it hard for the reader to follow. Additionally, ‘Satb2’ should be in italics in line 172.

We now annotate in line 168: (Fig. 2d, middle panels); and in line 170: (Fig. 2d, bottom panels). Also, “Satb2” is now italicized.

4) In figure 3B, the authors show that transcriptional priming of Satb2 using a flox-stop-flox-tdTomato plasmid electroporated in the Satb2Cre cortices. It would be helpful if the authors made it clearer in the text or the schematic in fig. 3A that in this model, Satb2 and Cre mRNA are expressed under the control of Satb2 promoter, whereas only Cre is translated but not Satb2, due to the differential protein translation outlined in fig. 2E. That is why Cre can be observed as soon as transcription takes place whereas Satb2 is absent.

We appreciate this great suggestion, and now explicitly emphasize the point in lines 209-212:

“Notably, both Satb2 and Cre mRNA are expressed under control of the same Satb2 promoter; however, we detect significantly fewer Satb2 protein-positive cells compared to Cre protein, further suggesting that Satb2 translation output is distinctly regulated (Fig. 3e).”

5) In figure 3E, the authors show protein expression for Cre, Satb2 and Bcl11b. I appreciate the added quantification of the immunostaining. However, no statistics were performed, and one

of the bars only has two datapoints. Again if possible, the dataset should be expanded and quantified.

*We have expanded quantification in **Figure 3e**, with now n^3 3 per experiment. We further perform statistical comparison of *Satb2* protein-positive cells, vs. *Cre*-positive or *Bcl11b*-positive cells. We include this description in the new **Figure 3e** legend and **Methods** sections, in addition to reporting the new values in **Supplementary Table 3**.*

6) In figure 6E, the immunostaining data presented is not very convincing and looks like background signal. This is also highlighted with discrepancy between low eIF4EBP1 P0 western blot signal but high eIF4EBP1 P0 immunostaining if the signal of eIF4EBP1 protein is to be believed. This should be clarified.

*We have improved the immunostaining for eIF4EBP1 in **Fig. 6e** – the original coronal section was obliquely cut, leading to more background signal when imaged in two dimensions. The new staining and imaging indeed reflect the Western blot.*

7) In figure 6G, the authors show a decrease in GFP+ cells in the cortical plate in eIF4EBP1 electroporations but quantifications do not reflect the image presented in figure 6F. A more representative image would help demonstrate the data more fairly.

*In new **Fig. 6g**, we present full-field eGFP and *Satb2* images, now also merged, in addition to zoomed images of the cortical plate vs. ventricular zone, to emphasize the difference quantified in new **Fig. 6h**. Fewer electroporated cells enter the cortical plate (CP; right panels under “zoom”) in the eIF4EBP1 shRNA condition in comparison to scrambled shRNA control.*

8) In figure 7C and lines 376-380, the clusters are mentioned out of order to the written texts. The authors are advised to rearrange the text according to the figure.

*We now rearrange the clusters in both the written text (lines 396-400), and GO pathways in **Fig. 7d**, to reflect the order in **Fig. 7c** for ease of reading and viewing.*

Reviewer #2:

Remarks to the Author:

Harnett and Ambrozkiwicz et al studied how gene expression changes during cortical development in the mouse, with an emphasis on post-transcriptional gene regulation. The authors perform a series of unbiased measurements to globally assess gene expression during mouse cortical development. By comparing datasets, the authors identify genes that are selectively translationally controlled. The authors find that *Satb2* is preferentially translated and ribosomal proteins are translationally repressed across development. The work is interesting and describing how gene expression is controlled during brain development is important. The multidisciplinary approach is fantastic and adds depth to the paper and demonstrates robustness of conclusions. Some lines of investigation stop short of exploring all the molecular details of the regulation identified, but that can be developed in future work as this is already a substantial contribution.

Major comments:

The introduction contains a large number of findings from this work in the last two paragraphs. In my opinion, these specific findings should be moved to the results or discussion sections so that the introduction can focus on setting the stage for the work.

*We appreciate this feedback, which led us to heavily edit and streamline the **Introduction**, where we now highlight only the most essential major points. Further details are reserved for the **Results**, and considered in depth in the expanded **Discussion**.*

The authors discuss the role of 4EBP1 at length, and also discuss 5' TOP dependent translational control, but don't discuss the mTOR signaling pathways that phosphorylate 4EBP1 to regulate 5' TOP translation. The authors also don't measure phosphorylation of 4EBP1 or other mTOR targets. In my opinion, this is an oversight in the work – especially since there is prior literature on the role of mTOR signaling in neuronal differentiation in human and mouse (e.g. PMID 29789464, 32115408, 29141229, 32876565, and more). At minimum, discussing this possibility would enrich the paper, and if the authors wanted to they could perform additional experiments to address these aspects. However, to be clear, this work is already very comprehensive and showing that 4EBP1 phosphorylation is important in this context is definitely beyond the scope of this manuscript.

The authors are particularly grateful for this suggestion, as it motivated us to investigate the phosphorylation of eIF4EBP1, leading to a more detailed understanding of its role in the timed downregulation of ribosomal protein translation.

*First, in new **Fig. 6d** we assess the developmental timing of eIF4EBP1 phosphorylation with a phosphospecific antibody probing Western blots of total neocortex lysates. We find that phosphorylation is most abundant at the earliest stages of neurogenesis, and declines sharply at E15.5. This decrease in phosphorylation occurs in advance of the decrease in eIF4EBP1 levels overall at E17. These data allowed us to rationalize how despite high levels of eIF4EBP1 at E12.5, ribosomal protein translation remains high, since phosphorylation leads to eIF4EBP1 dissociation from initiating complexes, thus dis-inhibiting 5'-TOP mRNA translation. Furthermore, the relative abundance of un-phosphorylated eIF4EBP1 at E15.5 would permit a sharp decrease in ribosomal protein translation at this stage, as reported in **Fig. 4b**, and again represented in **Fig. 6d**. Thus, this comment from Reviewer #2 led to a substantial refinement of our proposed mechanism, and a major improvement to the paper! These findings are now described in lines 329-337.*

Second, in new **Fig. 6f** we perform phospho-eIF4EBP1 immunohistochemistry in neocortex coronal sections corresponding to the time points analyzed for eIF4EBP1 overall in **Fig. 6e**. The results reinforce the Western blot findings, and further indicate that eIF4EBP1 phosphorylation occurs heavily in mitotic neural stem cells in the ventricular zone, and is retained throughout the ventricular zone and cortical plate at early neurogenesis E12.5. By E15.5, eIF4EBP1 phosphorylation declines particularly in the cortical plate, further rationalizing a mechanism for the sharp decline in ribosome abundance at this time and in this location (**Fig. 4c-d**). Interestingly, phosphorylation is nearly undetectable in progenitors of the glial lineage in the ventricular zone at P0, suggesting it is a particular feature of prenatal neurogenesis. These findings are added to the schematics in new **Fig. 6c and 6i**, and further described in lines 343-350.

Finally, we include in the **Discussion** a new reference, PMID 34272502²⁵, to connect eIF4EBP1 phosphorylation to mTOR, noting that our findings have implications for how neural stem cell regulation downstream of mTOR may act in the developing brain – lines 456-459:

“eIF4EBP1 is a master regulator of ribosome levels by suppressing ribosomal protein synthesis^{23,24}, playing a particularly dynamic role in stem cells downstream of mTOR²⁵, where we find it impacts the fate and migration of a neocortex neuronal lineage prenatally.”

A collection of mRNAs that are translationally controlled are identified in this work. The authors seek to then define what sequences in these mRNAs might be determinants of translation. The authors do this by identifying enriched sequence motifs in different sets of mRNAs, which is a strong analysis. However, this approach is unlikely to find some important regulatory elements, such as secondary structures and upstream start codons – both of which can be potent translational

regulators. The authors have generated a large amount of data and the paper would benefit from slightly deeper exploration of these features.

We now leverage our data to detect predicted secondary structures (G-quadruplexes) and translation activity upstream of the main open reading frame, comparing these 5'-UTR regulatory features in translation efficiency upregulated vs. downregulated mRNAs in new **Extended Data Fig. 8c-d**. We appreciate this suggestion from Reviewer #2, as it revealed interesting potential signatures of translational control during neocortex development:

We communicate these data to the reader in the **Discussion** (and **Methods**) to illustrate the utility of our Resource Article for the community to explore new regulatory features of translational control during neocortex development in lines 509-515:

“These data open new avenues for inquiry into gene expression regulation during neocortex development, such as the sequence determinants of translational control. For example, from E12.5 to E15.5, ribosome density in the 5'-UTR decreases for downregulated and increases for upregulated translation efficiency mRNAs (**Extended Data Fig. 8c**). Ribosome occupancy in the 5'-UTR may be indicative of upstream open reading frames (uORFs) and/or functional mRNA secondary structures. Indeed, we find that potential Gquadruplex-forming sequences are enriched in translation efficiency upregulated mRNAs (**Extended Data Fig. 8d**).”

Minor comments:

Ribosome profiling measures ribosome occupancy which may or may not be a direct measure of active translation since ribosomes can also stall, especially in neurons (e.g. PMID 24043809). The authors suggest that it is a direct measure of active translation, and clarification might improve the presentation of the technique.

*The authors appreciate that Reviewer #2 suggests highlighting the important distinction between “ribosome density” actually measured by ribosome profiling, vs. the term “translation efficiency” initially adopted by the field. Indeed, this was a point of much discussion during the preparation of the manuscript, and we initially opted to stick with the term “translation efficiency” used broadly in the field. We observe good correlation between ribosome density and protein concentration in our results, validating the common assumption that translation output for a given gene is usually proportional to ribosome flux. Now we take the opportunity to address this important point, and incorporate this great reference PMID 24043809⁷⁴ for the interpretation of ribosome profiling data in the **Discussion** lines 515-522:*

“Notably, while above we use the term “translation efficiency” broadly adopted by the field, Riboseq data is more accurately described as “ribosome density” on mRNA that may represent a wide range of phenomena from ribosome stalling to robust translation. Indeed, high ribosome density may reflect stalled polysomes poised for translation activation in response to synaptic activity, as previously described in cultured hippocampal neurons⁷⁴. The mechanistic significance of ribosome density on distinct sequence features in the neocortex transcriptome is an interesting direction for future study.”

Decision Letter, first revision:

Message: Our ref: NSMB-RS45687A

6th July 2022

Dear Dr. Kraushar,

Thank you for submitting your revised manuscript "A critical period of translational control during brain development at codon resolution" (NSMB-RS45687A). It has now been seen by one of the two original referees who has assessed both sets of responses to the two original referees. They find that the paper has improved in revision, and therefore we'll be happy in principle to publish it in Nature Structural & Molecular Biology, pending minor revisions to satisfy the referees' final requests and to comply with our editorial and formatting guidelines.

We are now performing detailed checks on your paper and will send you a checklist detailing our editorial and formatting requirements in about two weeks. Please do not upload the final materials and make any revisions until you receive this additional information from us.

To facilitate our work at this stage, we would appreciate if you could send us the main text

as a word file. Please make sure to copy the NSMB account (cc'ed above).

Sincerely,
Sara

Sara Osman, Ph.D.
Associate Editor
Nature Structural & Molecular Biology

Reviewer #2 (Remarks to the Author):

Exemplary reply to both reviewers - I am good with accepting the paper as is.

Decision Letter, author guidance:

Message: Our ref: NSMB-RS45687A

30th Aug 2022

Dear Dr. Kraushar,

Thank you for your patience as we've prepared the guidelines for final submission of your Nature Structural & Molecular Biology manuscript, "A critical period of translational control during brain development at codon resolution" (NSMB-RS45687A). Please carefully follow the step-by-step instructions provided in the attached file, and add a response in each row of the table to indicate the changes that you have made. Please also check and comment on any additional marked-up edits we have proposed within the text. Ensuring that each point is addressed will help to ensure that your revised manuscript can be swiftly handed over to our production team.

In recognition of the time and expertise our reviewers provide to Nature Structural & Molecular Biology's editorial process, we would like to formally acknowledge their contribution to the external peer review of your manuscript entitled "A critical period of translational control during brain development at codon resolution". For those reviewers

who give their assent, we will be publishing their names alongside the published article.

Nature Structural & Molecular Biology offers a Transparent Peer Review option for new original research manuscripts submitted after December 1st, 2019. As part of this initiative, we encourage our authors to support increased transparency into the peer review process by agreeing to have the reviewer comments, author rebuttal letters, and editorial decision letters published as a Supplementary item. When you submit your final files please clearly state in your cover letter whether or not you would like to participate in this initiative. Please note that failure to state your preference will result in delays in accepting your manuscript for publication.

Cover suggestions

As you prepare your final files we encourage you to consider whether you have any images or illustrations that may be appropriate for use on the cover of Nature Structural & Molecular Biology.

Nature Structural & Molecular Biology has now transitioned to a unified Rights Collection system which will allow our Author Services team to quickly and easily collect the rights and permissions required to publish your work. Approximately 10 days after your paper is formally accepted, you will receive an email in providing you with a link to complete the grant of rights. If your paper is eligible for Open Access, our Author Services team will also be in touch regarding any additional information that may be required to arrange payment for your article.

Please note that *Nature Structural & Molecular Biology* is a Transformative Journal (TJ). Authors may publish their research with us through the traditional subscription access route or make their paper immediately open access through payment of an article-processing charge (APC). Authors will not be required to make a final decision about access to their article until it has been accepted. [Find out more about Transformative Journals](https://www.springernature.com/gp/open-research/transformative-journals)

Authors may need to take specific actions to achieve [a](https://www.springernature.com/gp/open-research/funding/policy)

compliance-faqs"> compliance with funder and institutional open access mandates. If your research is supported by a funder that requires immediate open access (e.g. according to [Plan S principles](https://www.springernature.com/gp/open-research/plan-s-compliance)) then you should select the gold OA route, and we will direct you to the compliant route where possible. For authors selecting the subscription publication route, the journal's standard licensing terms will need to be accepted, including [self-archiving policies](https://www.nature.com/nature-portfolio/editorial-policies/self-archiving-and-license-to-publish). Those licensing terms will supersede any other terms that the author or any third party may assert apply to any version of the manuscript.

[Redacted]

Best regards,

Sophia Frank
Editorial Assistant
Nature Structural & Molecular Biology
nsmb@us.nature.com

On behalf of

Sara Osman, Ph.D.
Associate Editor
Nature Structural & Molecular Biology

Reviewer #2:
None

Reviewer #3:
Remarks to the Author:
Exemplary reply to both reviewers - I am good with accepting the paper as is.

Final Decision Letter:**Message** 19th Oct 2022

:

Dear Dr. Kraushar,

We are now happy to accept your revised paper "A critical period of translational control during brain development at codon resolution" for publication as an Article in Nature Structural & Molecular Biology.

Your paper will be published online soon after we receive proof corrections and will appear in print in the next available issue. You can find out your date of online publication by contacting the production team shortly after sending your proof corrections. Content is published online weekly on Mondays and Thursdays, and the embargo is set at 16:00

London time (GMT)/11:00 am US Eastern time (EST) on the day of publication. Now is the time to inform your Public Relations or Press Office about your paper, as they might be interested in promoting its publication. This will allow them time to prepare an accurate and satisfactory press release. Include your manuscript tracking number (NSMB-A45687B) and our journal name, which they will need when they contact our press office.

About one week before your paper is published online, we shall be distributing a press release to news organizations worldwide, which may very well include details of your work. We are happy for your institution or funding agency to prepare its own press release, but it must mention the embargo date and Nature Structural & Molecular Biology. If you or your Press Office have any enquiries in the meantime, please contact press@nature.com.

Please note that *Nature Structural & Molecular Biology* is a Transformative Journal (TJ). Authors may publish their research with us through the traditional subscription access route or make their paper immediately open access through payment of an article-processing charge (APC). Authors will not be required to make a final decision about access to their article until it has been accepted. [Find out more about Transformative Journals](https://www.springernature.com/gp/open-research/transformative-journals)

Authors may need to take specific actions to achieve [compliance](https://www.springernature.com/gp/open-research/funding/policy-compliance-faqs) with funder and institutional open access mandates. If your research is supported by a funder that requires immediate open access (e.g. according to [Plan S principles](https://www.springernature.com/gp/open-research/plan-s-compliance)) then you should select the gold OA route, and we will direct you to the compliant route where possible. For authors selecting the subscription publication route, the journal's standard licensing terms will need to be accepted, including [19](https://www.springernature.com/gp/open-research/policies/journal-

self-archiving policies. Those licensing terms will supersede any other terms that the author or any third party may assert apply to any version of the manuscript.

Sincerely,
Sara

Sara Osman, Ph.D.
Associate Editor
Nature Structural & Molecular Biology
